# Sumoylation regulates the assembly and activity of the SMN complex

Giulietta M. Riboldi [1,2,3,4,8], Irene Faravelli[1,2,3,8], Takaaki Kuwajima [1,2], Nicolas Delestrée[1,2,5], Georgia Dermentzaki[1,2], Mariangels De Planell-Saguer[1,2], Paola Rinchetti[1,2,3,6], Le Thi Hao[7], Christine C. Beattie[7,9], Stefania Corti [3,6], Serge Przedborski[1,2,5], George Z. Mentis[1,2,5] & Francesco Lotti [1,2,5]✉

SMN is a ubiquitously expressed protein and is essential for life. SMN deficiency causes the neurodegenerative disease spinal muscular atrophy (SMA), the leading genetic cause of infant mortality. SMN interacts with itself and other proteins to form a complex that functions in the assembly of ribonucleoproteins. SMN is modified by SUMO (Small Ubiquitin-like Modifier), but whether sumoylation is required for the functions of SMN that are relevant to SMA pathogenesis is not known. Here, we show that inactivation of a SUMO-interacting motif (SIM) alters SMN sub-cellular distribution, the integrity of its complex, and its function in small nuclear ribonucleoproteins biogenesis. Expression of a SIM-inactivated mutant of SMN in a mouse model of SMA slightly extends survival rate with limited and transient correction of motor deficits. Remarkably, although SIM-inactivated SMN attenuates motor neuron loss and improves neuromuscular junction synapses, it fails to prevent the loss of sensory-motor synapses. These findings suggest that sumoylation is important for proper assembly and function of the SMN complex and that loss of this post-translational modification impairs the ability of SMN to correct selective deficits in the sensory-motor circuit of SMA mice.

[1] Center for Motor Neuron Biology and Disease, Columbia University, New York, NY, USA. [2] Department of Pathology and Cell Biology, Columbia University, New York, NY, USA. [3] Dino Ferrari Centre, Neuroscience Section, Department of Pathophysiology and Transplantation (DEPT), University of Milan, Milan, Italy. [4] The Marlene and Paolo Fresco Institute for Parkinson's and Movement Disorders, NYU Langone Health, New York, USA. [5] Department of Neurology, Columbia University, New York, NY, USA. [6] Foundation IRCCS Ca' Granda Ospedale Maggiore Policlinico, Neurology Unit, Milan, Italy. [7] Department of Neuroscience, Ohio State University, Columbus, OH, USA. [8] These authors contributed equally: Giulietta M. Riboldi, Irene Faravelli. [9] Deceased: Christine C. Beattie. ✉email: fl2219@cumc.columbia.edu

Survival of motor neuron (SMN) is a ubiquitously expressed protein that localizes to both the cytoplasm and the nucleus, where it accumulates in distinct nuclear bodies termed Gems and Cajal bodies[1,2]. SMN associates with itself and at least eight additional proteins (GEMIN2-8 and UNRIP) to form the SMN complex[3,4]. This multimeric complex has a well-characterized role in the biogenesis of spliceosomal small nuclear ribonucleoproteins (snRNPs)[5,6] as well as U7 snRNP that functions in the 3′ end processing of histone mRNAs[7,8]. SMN has also been implicated in other aspects of RNA regulation including the assembly of messenger ribonucleoprotein (mRNP) complexes[9]. Consistent with its central role in RNA processing[4,10], SMN deficiency has been shown to induce widespread splicing dysregulation and transcriptome alterations in a variety of in vivo models[11–15].

*SMN1* gene mutations or deletions result in spinal muscular atrophy (SMA), a devastating neurodegenerative disorder characterized by the progressive loss of motor neurons and skeletal muscle atrophy[16,17]. In addition to motor neuron loss, characterization of SMA pathogenesis using animal models has identified multiple perturbations in the motor circuit that include dysfunction and loss of neuromuscular junctions (NMJs) and central proprioceptive sensory synapses onto motor neurons[18,19]. Previously, we have shown that motor neuron death in a mouse model of severe SMA is mediated by activation of the tumor suppressor p53[20], while sensory-motor circuit impairments are driven by dysfunction of *Stasimon* (*Stas*), a U12 intron-containing gene regulated by SMN[21,22].

The multifunctional nature of the SMN complex requires its interaction with many different protein partners, but how the specificity of these protein-protein interactions is achieved is not known. Post-translational modifications (PTMs) of proteins can regulate complex and dynamic cellular processes by facilitating interactions between key proteins[23]. Small ubiquitin-like modifier (SUMO) conjugation (or sumoylation) has gained prominence as a regulatory mechanism of a variety of biological processes and has been recently linked to SMN function in cells[24–26]. Chief among the different biological processes is the role that sumoylation plays in regulating nucleocytoplasmic transport both through steric modifications of the cargo molecules and also by directly interacting with units of the nuclear transport machinery[27]. Reports have identified a series of proteins that are not retained in the nucleus once sumoylation is inhibited, reinforcing the hypothesis that sumoylation can be essential to prevent nuclear export[28–30]. Conversely, other observations support a role of sumoylation in stimulating nuclear export[31,32]. SUMO modification has also been shown to promote the nuclear import of polo-like kinase 1 (PLK1) and to prevents its proteasomal degradation[33].

Sumoylation involves the covalent attachment of SUMO proteins to lysine residues on target proteins, and is mediated by activating (E1), conjugating (E2), and ligating (E3) enzymes[34–37]. UBC9 is the only E2 enzyme that mediates SUMO transfer to substrate proteins[38,39]. Sumoylation is dynamically regulated by a family of SUMO-specific proteases that catalyze SUMO deconjugation[36,40]. Mammalian cells possess three SUMO isoforms that can be covalently conjugated to proteins as a single moiety (SUMO-1) or as polymeric SUMO chains (SUMO-2 and SUMO-3)[37]. In addition to the covalent attachment of SUMO to substrates, a growing number of proteins, including SMN[26], have been reported to bind SUMO non-covalently via SUMO interaction motifs (SIMs). Interactions between SUMO and SIM-containing proteins are essential regulators of nuclear bodies dynamics as they foster physical interactions between proteins through SUMO-SIM networks[41–43]. Interestingly, loss of the SIM motif in SMN impairs its interaction with an essential component of snRNPs and Cajal bodies[26]. However, the functional consequences of losing interaction between SMN and SUMO and their potential contribution to SMA pathogenesis have not been investigated.

In this study, we functionally characterized the role of the SIM motif of SMN in cellular and animal models of SMA. We found that the SIM motif of SMN is crucial for the assembly, localization, and stability of the SMN complex. Importantly, we found that loss of the SIM motif severely impairs SMN function in snRNP biogenesis. Moreover, the expression of a SIM-inactivated mutant of SMN fails to prevent motor neuron development defects induced by SMN deficiency in zebrafish. In SMA mice, expression of SIM-inactivated SMN modulates motor neuron death and NMJ denervation, while failing to rescue sensory-motor connectivity deficits. At the molecular level, SIM-inactivated SMN prevents the upregulation of p53 transcriptional targets, while failing to correct *Stas* splicing. Together, these results establish SUMO modification as an important determinant of the integrity and function of the SMN complex and link this PTM to selective aspects of sensory-motor circuit dysfunction in SMA models.

## Results

**Sumoylation is required for the localization and integrity of the SMN complex.** To investigate the effects of reduced sumoylation on SMN complex function, we used lentiviral vectors to generate a stable cell line with a doxycycline (Dox)-inducible, RNAi-mediated knockdown of the SUMO-conjugating enzyme UBC9 (Supplementary Fig. 1a). HeLa-UBC9$_{RNAi}$ cells cultured in the presence of Dox for 4 days showed a strong reduction of UBC9 mRNA (Supplementary Fig. 1b) and protein levels compared to untreated cells (Fig. 1a). As expected, UBC9 deficiency severely decreased SUMO conjugation as demonstrated by Western blot analysis of SUMO-2/3 immunoprecipitations (Fig. 1b). Importantly, impaired SUMO conjugation altered SMN complex subcellular distribution, reducing SMN localization in nuclear gems and aberrant accumulation in cytoplasmic foci (Fig. 1c). Aside from SMN, other core components of the SMN complex, including GEMIN2, GEMIN3, GEMIN5, and GEMIN6 were mislocalized in UBC9-deficient cells, suggesting that the entire SMN complex accumulates in the cytoplasm when sumoylation is inhibited (Fig. 1d and Supplementary Fig. 1c). Given that sumoylation can regulate nucleocytoplasmic trafficking of proteins[27], we asked whether UBC9 depletion impairs SMN nuclear import. We performed nucleocytoplasmic fractionation of HeLa-UBC9$_{RNAi}$ cells cultured in the absence or presence of Dox for 4 days. We found that inhibition of sumoylation results in a redistribution of SMN from the nucleus to the cytoplasm (Supplementary Fig. 1d and 1e), suggesting that the altered localization of SMN observed in these cells upon UBC9 depletion is due to an impaired nuclear transport.

To investigate the effects of reduced sumoylation on the integrity of the SMN complex, lysates from HeLa-UBC9$_{RNAi}$ cells cultured for 4 days in the absence or presence of Dox were fractionated by 5–20% sucrose-gradient sedimentation and analyzed by Western blot. As previously published[44,45], SMN sediments in a broad 40–70 S peak centered in fractions 10–14 (Fig. 1e). Following UBC9 knockdown, however, SMN could be also detected in lighter fractions, suggesting that sumoylation is required for the proper formation of the SMN complex. In line with this possibility, we found that SMN interaction with GEMIN3 and GEMIN5 is weakened by inhibition of sumoylation (Supplementary Fig. 1f). Overall, these results show that sumoylation is required for SMN complex localization and integrity in human cells.

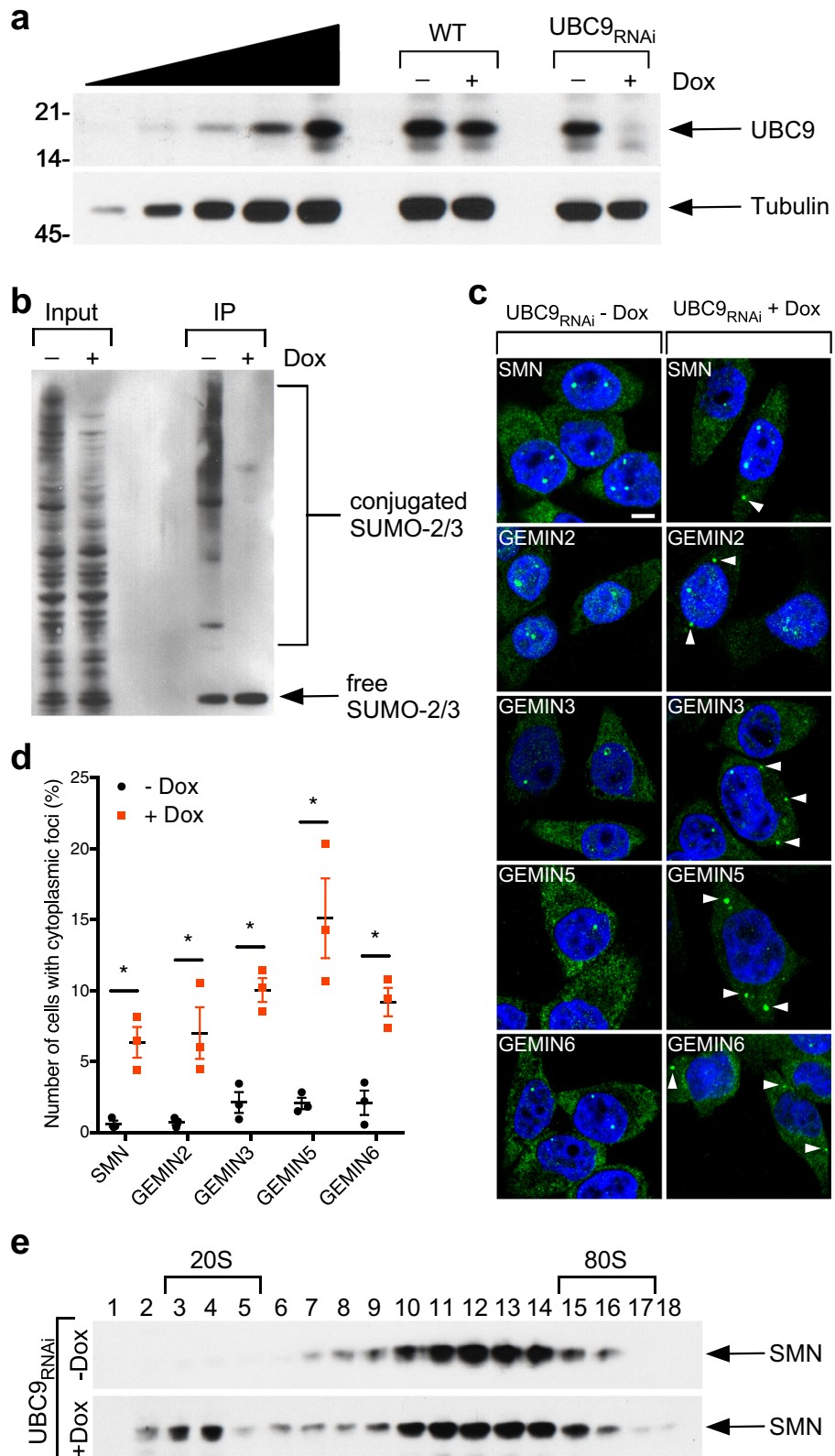

**Components of the SMN complex are modified by SUMO.** Given the mislocalization of the SMN complex caused by the loss of sumoylation, we sought to investigate the role of this PTM on the different components of the complex. Although SMN has been previously reported to be a substrate for SUMO-1[26], impaired sumoylation also altered the subcellular location of other subunits of the SMN complex, suggesting that other components of the complex might be sumoylated. Covalent sumoylation often occurs within a consensus sequence ΨKXE/D—where Ψ is a hydrophobic residue, K is the lysine conjugated to SUMO, X represents any amino acid, D or E is an acidic residue[46]. Using three independent software—SUMOplot[TM], JASSA[47], and GPS-SUMO[48]—we generated a map of all predicted SUMO acceptor sites of all the core components of the SMN complex. We found sumoylation sites with a high probability of

**Fig. 1 Sumoylation is required for the localization and integrity of the SMN complex. a** SDS-PAGE and Western blot analysis of endogenous UBC9 protein levels in HeLa wildtype (WT) or UBC9$_{RNAi}$ cells cultured without (-) or with (+) Doxycycline (Dox) for 4 days. A twofold serial dilution of WT HeLa cell extract is analyzed on the left. **b** Western blot analysis of anti-SUMO-2/3 immunoprecipitates in HeLa WT or UBC9$_{RNAi}$ cells cultured without (−) or with (+) Dox for 4 days. **c** Representative fluorescent images of HeLa UBC9$_{RNAi}$ cells cultured without (−) or with (+) Dox for 4 days and stained with antibodies against SMN, GEMIN2, GEMIN3, GEMIN5, and GEMIN6 (green), and with DAPI (blue). Cytoplasmic foci are indicated with white arrowheads. Scale bar, 10 μm (figures are representative of $n = 3$ biologically independent experiments). **d** Quantification of number of cells with cytoplasmic foci from the same groups as in (**c**). Data represent means and SEM ($n = 3$ biologically independent experiments). Statistical significance was determined by two-sided, unpaired multiple $t$ tests followed by Holm-Sidak correction (adjusted $P$ values −Dox vs. +Dox: SMN = 0.023007; GEMIN2 = 0.026589; GEMIN3 = 0.010270; GEMIN5 = 0.023007; GEMIN6 = 0.023007). **e** Cell extracts from HeLa UBC9$_{RNAi}$ cells cultured without (−) or with (+) Dox for 4 days were fractionated by 10–30% sucrose gradient centrifugation. Endogenous SMN levels in each fraction were analyzed by SDS-PAGE and Western blot. Numbers of fractions and sedimentation (S) values are indicated.

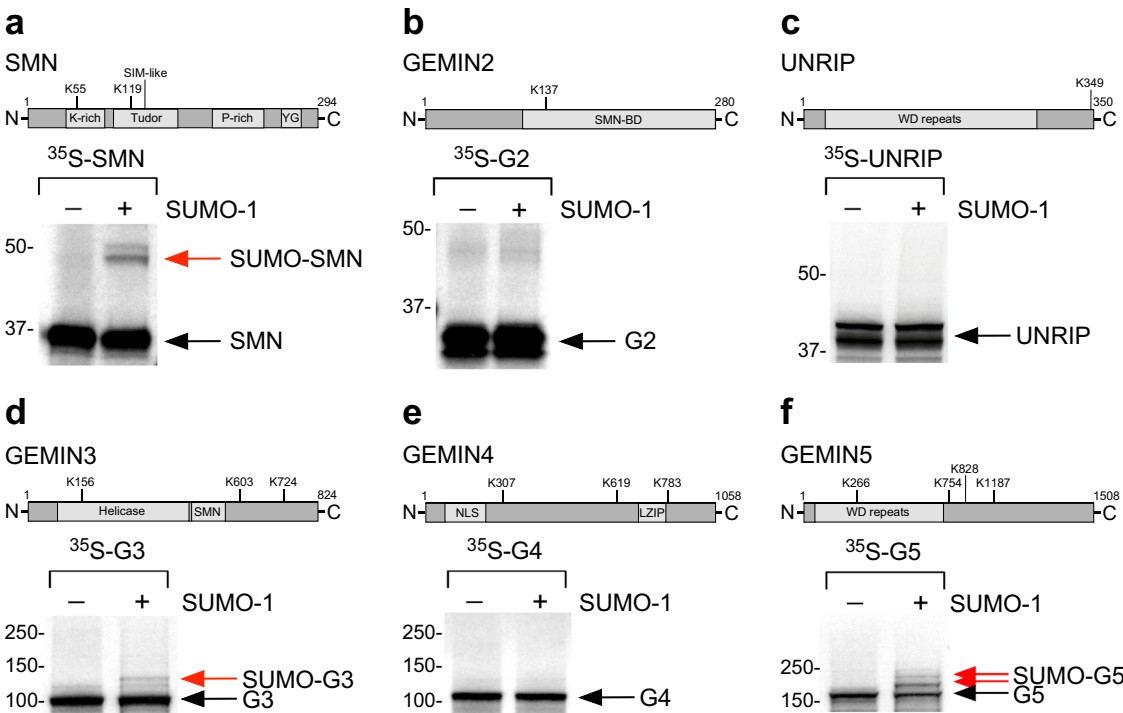

**Fig. 2 Components of the SMN complex are modified by SUMO.** Schematic representation of the bioinformatically predicted sumoylation sites in SMN (**a**), GEMIN2 (**b**), UNRIP (**c**), GEMIN3 (**d**), GEMIN4 (**e**), GEMIN5 (**f**) and in vitro sumoylation of the respective $^{35}$S-labeled recombinant proteins. Red arrows indicate the position of the SUMO-modified proteins.

being modified in SMN, GEMIN2, GEMIN3, GEMIN4, GEMIN5, and UNRIP (Supplementary Table 1). To validate these bioinformatics predictions, we used an in vitro sumoylation assay in which $^{35}$S-labeled proteins are incubated with components of the SUMO pathway either in the presence or absence of ATP[49]. We found that in addition to SMN, GEMIN3 and GEMIN5 are modified by SUMO-1, while GEMIN2, GEMIN4, and UNRIP are not (Fig. 2). In addition to modification by SUMO-1, SMN, GEMIN3, and GEMIN5 are also modified by SUMO-2 (Supplementary Fig. 2). A review of available proteomics data on sumoylated proteins confirmed the identification of SMN and GEMIN5 as sumoylated targets, with the latter showing an overlap with the bioinformatically predicted site on lysine 754 (Supplementary Table 2)[50,51]. While these results explain why the loss of sumoylation affects not only SMN but the entire complex, they also suggest that sumoylation can act to foster binding between core components of the SMN complex through a network of SUMO-SIM interactions.

**The SIM domain of SMN is required for its interaction with SUMO modifiable components of the SMN complex.** In addition to covalent SUMO modification motifs, SMN contains a

putative SUMO-interacting motif (SIM) within its Tudor domain that could mediate interaction with SUMO-modified proteins (Fig. 3a)[26]. The Tudor domain within SMN facilitates protein-protein interactions with several binding partners[52]. This domain binds to the C-terminal tails of Sm proteins which contain symmetrical dimethylated arginine residues, thereby facilitating their assembly on U snRNAs[53,54]. The presence of a SIM within the SMN Tudor domain may provide a different modality for the interaction of SMN with SUMO modified core components of the SMN complex as well as other protein substrates. Indeed, SIM-less mutants of SMN have been reported to interact less efficiently with SmD1, a component of the heptameric Sm core that the SMN complex assembles on spliceosomal U snRNAs[26]. However, a clear demonstration that the SIM domain of SMN is binding SUMO has not yet been reported. To prove that SMN requires this SIM motif to interact with SUMO, we generated a SIM-inactivated mutant of SMN by changing two valine residues to alanine (SMN-V124-125A, hereafter called SMN-2VA, Fig. 3a). We generated recombinant WT and 2 VA proteins spanning the Tudor domain of SMN and used them to pull-down His-tagged SUMO-1 or SUMO-2 recombinant proteins. The result showed that SMN interacts with SUMO-2 and to a lesser extent with

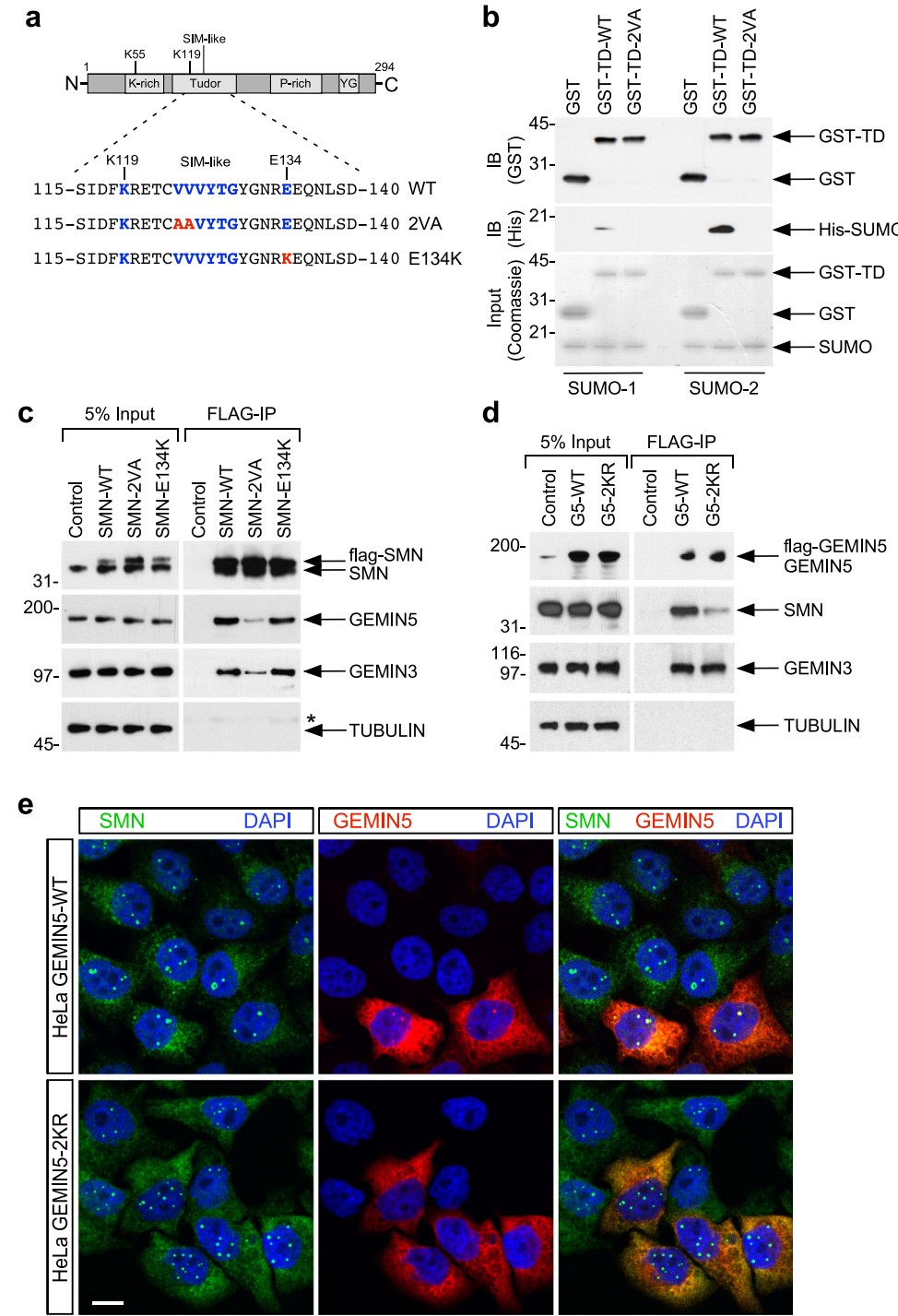

SUMO-1 and that this interaction is lost by mutations in the SIM motif of SMN (Fig. 3b).

To determine whether the SIM domain of SMN is required for binding to sumoylatable Gemins, we expressed FLAG-tagged WT and SIM-inactivated SMN in HEK293T cells and purified the SMN complex using FLAG-M2 agarose beads[6]. Western blot analysis of the purified SMN complexes revealed impaired SMN-2VA interaction with endogenous GEMIN5 and GEMIN3 (Fig. 3c). Interestingly, SMN-E134K, a mutation in the Tudor domain that causes SMA, does not affect SMN interaction with other Gemins (Fig. 3c), despite the fact that E134K mutation induces structural instability of the Tudor domain and disrupts SMN nuclear import[55,56]. To confirm that the observed

impairment is the result of SIM-inactivated SMN inability to interact with the SUMO moiety of Gemins, we generated SUMO-deficient mutant of GEMIN5. Bioinformatics prediction reports four main sumoylation sites for GEMIN5 (Supplementary Fig. 3a). To identify which of the potential SUMO sites of GEMIN5 are functional, we mutated each one individually or in combination, and examined sumoylation in vitro. Single point mutations of either K266 (K1) or K1187 (K4) had no effect on GEMIN5 sumoylation (Supplementary Fig. 3a) while mutations in K754 (K2) or K828 (K3) caused a partial loss of sumoylation. Concomitant mutation of both K754 and K828 residues (K2-3) resulted in the complete loss of GEMIN5 sumoylation (Supplementary Fig. 3b). We then expressed a FLAG-tagged WT and a

**Fig. 3 The SIM domain of SMN is required for its interaction with SUMO-modifiable components of the SMN complex. a** Schematic representation of the different SMN protein domains and amino acid residues in the portion of the Tudor domain that contains the SUMO-interacting motif (SIM). The residue positions of the SIM are indicated in blue. K119 is the putative sumoylation site based on SUMOplot analysis. E134 is the position of a mutation found in SMA patients (E134K). The changes in SMN protein sequence to generate SMN-2VA and SMN-E134K mutants are reported in red. **b** The SIM-like motif of SMN is required to directly bind with SUMO in vitro. Equal protein amounts of purified His-SUMO1 or SUMO-2 (Input) were individually incubated with wildtype (WT) SMN Tudor domain (WT-TD) or its SIM mutants (TD-2VA) of GST-fusion proteins purified from *E. Coli* expression system and pulled down with Glutathione-sepharose beads. Bound complexes were analyzed by immunoblotting (IB) with the indicated antibodies. The amounts of recombinant proteins were evaluated by Coomassie staining of 10% of the input. **c** HEK-293T cells were transfected with FLAG-tag SMN wildtype (WT), SIM-less (2 VA), or SMA (E134K) mutants. Cell lysates were prepared for precipitation with anti-FLAG agarose beads. 5% of the input and the immunoprecipitates (FLAG-IP) were analyzed by SDS-PAGE and Western blot with antibodies against the proteins indicated on the right. Naïve HEK-293T cells were used as control. * indicate IgG heavy chain. **d** HEK-293T cells were transfected with FLAG-tag GEMIN5 wildtype (G5-WT) or its non-sumoylatable mutant (G5-2KR). Cell lysates were prepared for precipitation with anti-FLAG agarose beads. Five percent of the input and the immunoprecipitates (FLAG-IP) were analyzed by SDS-PAGE and Western blot with antibodies against the proteins indicated on the right. Naïve HEK-293T cells were used as control. **e** Representative fluorescent images of HeLa cells transfected with wildtype (WT) or non-sumoylatable (2KR) GEMIN5 and stained with antibodies against SMN (green) and FLAG (GEMIN5 in red), and with DAPI (blue). Scale bar, 20 μm.

non-sumoylatable mutant of GEMIN5 (K2-3, hereafter called G5-2KR) in HEK293T cells, and purified the GEMIN5-binding proteins using FLAG-M2 agarose beads. Western blot analysis showed that GEMIN5 interaction with SMN is reduced by the G5-2KR mutant (Fig. 3d). Importantly, impaired GEMIN5 sumoylation altered its subcellular distribution, abolishing its localization in gems and its interaction with SMN in these nuclear structures (Fig. 3e). Altogether, these results indicate that GEMIN5 sumoylation is required for binding to SMN through its SIM domain.

**Loss of SMN SIM domain alters the integrity and localization of the SMN complex and its function in the assembly of snRNPs.** To investigate the role of the SIM domain on SMN complex integrity and localization, we used a previously established cell line (NIH3T3-Smn$_{RNAi}$) with Dox-inducible RNAi-mediated knockdown of endogenous SMN in which SMN depletion decreases snRNP assembly, reduces snRNP levels, and induces RNA splicing defects, as well as arresting cell growth[21]. Importantly, all molecular and phenotypic defects in these cells can be corrected by transgenic expression of wild-type (WT) RNAi-resistant human SMN[21]. We then generated NIH3T3-Smn$_{RNAi}$ cell lines stably expressing either SMN-WT or SMN-2VA (Supplementary Fig. 4a). To control for uneven transduction, we measured the expression levels of WT and mutant SMN transcripts by RT-qPCR and found that WT and 2 VA SMN mRNAs are expressed at comparable levels (Supplementary Fig. 4b). Importantly, as in the parental NIH3T3-Smn$_{RNAi}$ cell line, each of the new cell lines maintained the capacity for RNAi-mediated downregulation of endogenous Smn in the presence of Dox (Supplementary Fig. 4c and 4d). Comparison at the protein levels showed that SMN-2VA steady-state levels are slightly less than SMN-WT levels in the presence of endogenous SMN (−Dox) and that SMN-2VA levels are further reduced upon induction of endogenous SMN silencing (+Dox) (Supplementary Fig. 4d and 4e).

Employing the described cellular model, we first sought to determine the effects of loss of SMN SIM domain on the integrity of the SMN complex. We performed sucrose-gradient fractionation followed by Western blot analysis of endogenous Smn-depleted NIH3T3-Smn$_{RNAi}$ cells expressing either SMN-WT or SMN-2VA. Strikingly, mutation of the SIM domain in SMN impairs its ability to form large macromolecular complexes suggesting that SUMO-SMN interactions are required for the structural integrity of the SMN complex (Fig. 4a). Concordantly, immunoprecipitations of cell lysates from Smn-depleted NIH3T3-Smn$_{RNAi}$ cells expressing either SMN-WT or SMN-2VA show that the binding of SMN to core components of the

SMN complex is impaired by mutation of the SIM domain (Supplementary Fig. 4f).

To investigate the role of SIM in the integrity of the subcellular localization of the complex, we performed SMN immunostaining using anti-FLAG tag antibodies, which selectively recognize transgenic human SMN in endogenous Smn-depleted NIH3T3-Smn$_{RNAi}$ cells expressing either SMN-WT or SMN-2VA. Strikingly, immunofluorescence analysis showed that loss of SMN interaction with SUMO does not cause aberrant cytoplasmic foci of SMN, but causes an almost complete loss of nuclear gems (Fig. 4b and c). Interestingly, loss of SMN-SUMO interaction does not affect its nuclear import as demonstrated by nucleocytoplasmic fractionation of NIH3T3-Smn$_{RNAi}$ cells expressing either SMN-WT or SMN-2VA (Supplementary Fig. 4g and h), consistent with what previously reported[26].

Next, we asked whether loss of SMN SIM domain influences its protein stability, as suggested by reduced steady-state levels of SMN-2VA (Supplementary Fig. 4d and 4e). We inhibited protein synthesis with cycloheximide (CHX 100 μg/ml) and analyzed SMN protein levels by Western blot in NIH3T3-Smn$_{RNAi}$ cells expressing WT or 2 VA SMN and treated with both doxycycline and CHX. Western blot using anti-Strep tag antibodies revealed that SMN-2VA has a shorter half-life than SMN-WT (Fig. 4d and e). Thus, the loss of SUMO-SIM interaction increases the turnover of SMN. Overall, these results indicate that SUMO-SIM binding is important for SMN complex localization and stability.

The best-characterized function of the SMN complex is in the assembly of spliceosomal and U7 snRNPs[6,8]. To investigate whether sumoylation is required for SMN function in snRNP biogenesis, we performed snRNP assembly assays with radioactively labeled U1 snRNA, followed by immunoprecipitation with anti-Sm antibodies[6,57,58]. Importantly, we found that SMN-2VA has an impaired snRNP assembly activity (Fig. 4f and g). Together, these results indicate that sumoylation is required for both the integrity and the function of the SMN complex.

**SIM-less SMN fails to rescue motor axon deficits in a zebrafish model of SMA.** To determine the effects of SIM loss on SMN function in vivo, we utilized a zebrafish model of SMA. Maternalzygotic SMN (*mz-smn*) mutant of zebrafish is a genetic model that results from depletion of *smn* from the earliest stages of development by removing both maternal and zygotic *smn*. *mz-smn* mutants show defects in motor axon outgrowth and motor neuron development[59]. We set out to test whether SIM-inactivated SMN fails to rescue these SMN-dependent defects in vivo.

We injected RNAs coding for human SMN-WT or SMN-2VA into the *mz-smn* mutants. Western blot analysis shows that both

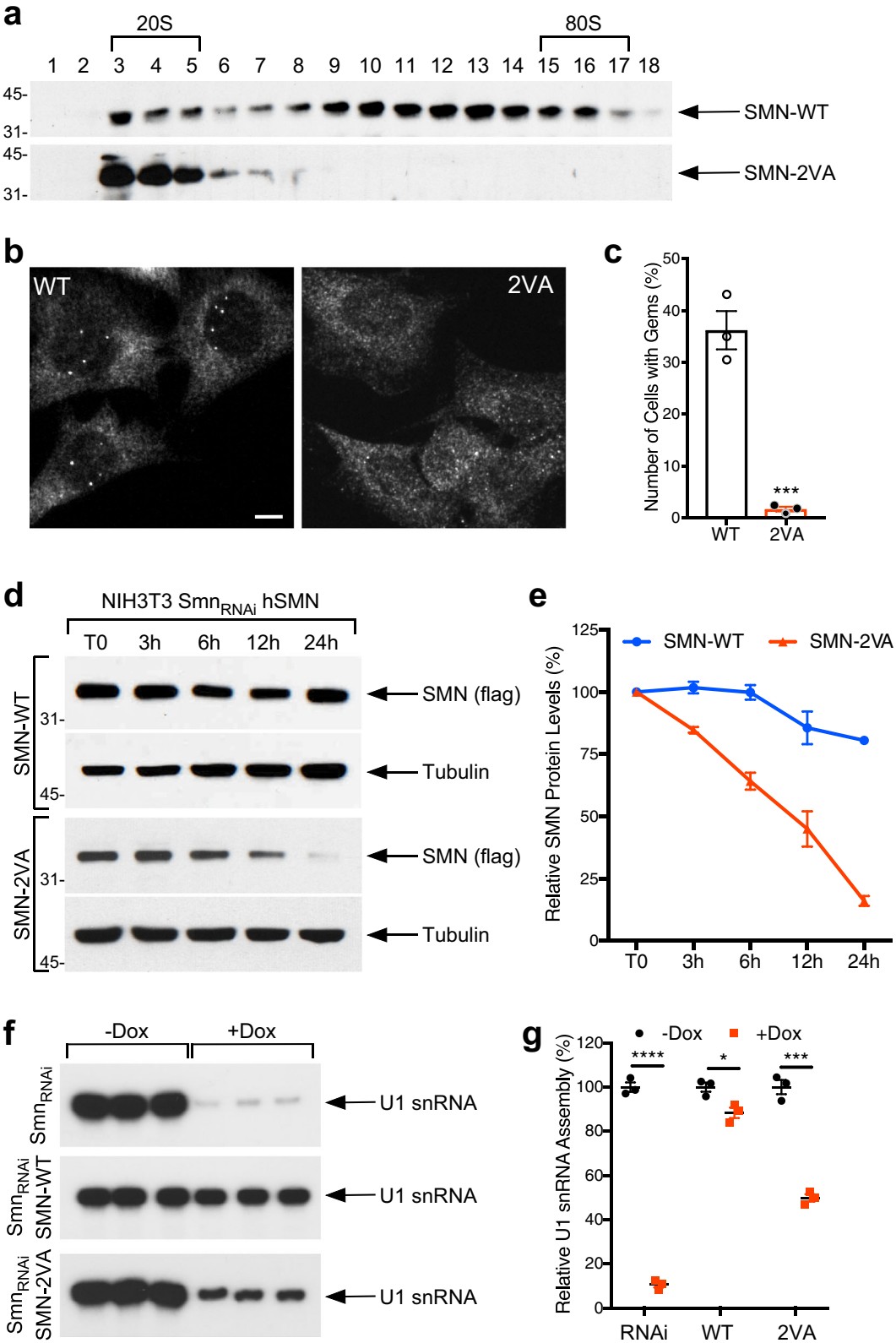

WT and mutant SMN protein are expressed at comparable levels (Fig. 5a). Animals were scored as having severe, moderate, mild or no defects of their motor axons 28 h post fertilization (hpf) as previously described[60]. When compared with controls, motor axons from severely affected embryos are characterized by multiple truncations with or without branching. On the other hand, moderately affected embryos present a wider range of

alterations, spanning from multiple axons innervating the neighboring myotome, to a single truncation.

As shown in Fig. 5b and quantified in Fig. 5c, *mz-smn* mutants have abnormal motor axon morphologies and WT human SMN rescued this abnormal phenotype as previously reported[21,59,60]. Interestingly, the SIM-inactivated SMN mutant was unable to restore motor neuron outgrowth and displayed a severe

**Fig. 4 Loss of SMN SIM domain alters the localization and stability of the SMN complex and its function in the assembly of small nuclear ribonucleoproteins. a** Cell extracts from NIH3T3-Smn$_{RNAi}$ cells expressing SMN-WT or SMN-2VA and treated with doxycycline (Dox) for 7 days were fractionated by 10-30% sucrose gradient centrifugation. SMN in each fraction was detected by SDS-PAGE and Western analysis. Number of fractions and sedimentation (S) values are indicated. **b** Representative images of NIH3T3-Smn$_{RNAi}$ cells expressing SMN-WT or SMN-2VA and cultured with Dox for 7 days and stained with anti-flag antibody. Scale bar, 10 µm. **c** Quantification of the number of cells with nuclear Gems in NIH3T3-Smn$_{RNAi}$ cells expressing SMN-WT or SMN-2VA and cultured with Dox for 7 days. Data represent means and SEM of three independent experiments ($n = 3$). Statistical significance was determined by two-tailed unpaired $t$ test ($P$ value = 0.0007). **d** Representative western blot analysis of SMN levels in NIH3T3-Smn$_{RNAi}$ cells expressing SMN-WT or SMN-2VA and treated with Dox for 7 days and cycloheximide at different time points. Tubulin is used as loading control. **e** Quantification of the SMN levels from three independent Western blots experiments as in (**d**) ($n = 3$ independent biological experiments). Data represent mean and SEM. **f** Equal amounts of cell extracts from NIH3T3-Smn$_{RNAi}$ cells expressing SMN-WT or SMN-2VA and treated without (−) or with (+) Dox for 7 days were analyzed in snRNP assembly reactions by immunoprecipitation of radioactive U1 snRNA with anti-SmB (18F6) antibody followed by electrophoresis on denaturing polyacrylamide gels and autoradiography. **g** The amount of U1 snRNA immunoprecipitated in snRNP assembly experiments as in (**f**) was quantified using a Typhoon Phosphorimager and expressed as a percentage of that in samples without Dox treatment. Data represent means and SEM of three independent biological experiments ($n = 3$). Statistical significance was determined by two-sided, unpaired multiple $t$ tests followed by Holm-Sidak correction (adjusted $P$ values −Dox vs. +Dox: RNAi = 0.000011; WT = 0.020757; 2 VA0.000319).

phenotype close to that of *mz-smn* mutants (Figs. 5b and 5c). These data imply that SUMO-SIM interactions are important for SMN function in zebrafish motor neuron development.

**SIM-inactivated SMN gene delivery fails to rescue survival and motor function in SMA mice.** Extensive work in mouse models of SMA has established that pathology is not restricted to motor neurons, but also involves impairment of the entire sensory-motor system[21,61]. Thus, SMA is likely not caused by disruption of a single SMN-dependent RNA pathway in motor neurons, but rather by disruption of multiple pathways across the entire motor circuit[17,62]. Adeno-associated virus (AAV)-mediated delivery of SMN leads to a striking phenotypic correction in severe SMA mice[8,63,64]. AAV9-mediated gene delivery could be employed to assess the effect of mutations of SMN based on their ability to rescue the SMA phenotype in mouse models. Accordingly, we used this approach to investigate the effect of loss of SMN sumoylation in mouse models of SMA as a means to uncover which and how downstream events disrupted by SMN deficiency contribute to the disease phenotype.

First, we cloned human SMN WT and SIM-inactivated mutant (SMN-2VA) into an AAV9 vector under the control of the ubiquitous beta-glucuronidase (GUSB) promoter (Supplementary Fig. 5a). The new constructs were then used for custom production of high-titer viral preparations for injection in SMN-Δ7 SMA mice. In addition to AAV9-SMN-2VA, the scAAV9-GFP and scAAV9-SMN-WT vectors were used as negative and positive controls, respectively. Equal amounts of viral genomes (~1 × 10$^{11}$ gc per mouse) were injected into the cerebral lateral ventricle (ICV) of SMN-Δ7 SMA mice at birth (P0). The efficiency of AAV9-mediated targeting of SMA-relevant sensory-motor circuit neurons was confirmed by monitoring GFP expression by immunohistochemistry, which resulted in high transduction of motor neurons in the spinal cord and sensory neurons in dorsal root ganglia (DRG) as previously reported[22,65] (Supplementary Fig. 5b–d). RNA and protein analyses of spinal cord protein lysates showed that SMN was expressed at a comparable level in the injected mice (Fig. 6a–c).

We then monitored motor behavior (righting reflex), weight gain, and survival of SMN-injected SMA mice relative to GFP-treated mice (Fig. 6d-f). Importantly, mice expressing SMN-2VA survived slightly longer (19-days median) than mice expressing GFP (12-days median), but dramatically less than those expressing SMN-WT (98-days median, Fig. 6d). Righting reflex and weight gain followed the same pattern, with SMN-2VA treated mice transiently improving during the first 10 days after injection and quickly deteriorating thereafter (Fig. 6e and f). Altogether, the phenotype of SMN-2VA treated mice was

improved in comparison with GFP injected animals, but this beneficial phenotypic effect appeared to be transient and it did not improve the disease phenotype to the levels of WT-SMN treated mice.

**SIM-inactivated SMN corrects motor neuron loss and partially prevents NMJ denervation in SMA mice.** Spinal motor neuron loss is a hallmark of SMA pathology[16,17,66]. Besides this motor neuron death, SMA pathology is characterized by multiple synaptic deficits in the motor circuit that include dysfunction and loss of NMJs and proprioceptive sensory synapses onto motor neurons[18,61,67,68]. To test whether SIM-inactivated SMN corrected any of these abnormalities, we injected SMN-Δ7 SMA mice at P0 either with AAV-GFP, AAV-SMN-WT or AAV-SMN-2VA and at P9 performed a comprehensive panel of anatomical and functional assays to assess the changes in the spinal sensory-motor circuit.

To investigate whether SIM-inactivated SMN could correct the loss of SMA spinal motor neurons, we used antibodies against Choline Acetyltransferase (ChAT) to visualize and count motor neurons in the vulnerable lumbar 1 (L1) segment of the spinal cord (Fig. 7a). SMA mice treated with AAV-GFP displayed a loss of approximately 60% of L1 motor neurons at P9 compared to WT mice that were partially corrected by SMN-WT expression (Fig. 7c). Remarkably, SMN-2VA expression was able to correct this loss of SMA motor neurons to levels similar to the SMN-WT expression (Fig. 7c).

To assess the effect of SIM-inactivated SMN expression on NMJ innervation, we used antibodies against neurofilament (NF-M) and synaptophysin (SYP) as presynaptic NMJ markers and fluorescently-labeled α-bungarotoxin (BTX) to label the post-synaptic acetylcholine receptor clusters on muscle fibers. The percentage of fully denervated NMJs was evaluated considering the colocalization of the pre- and post-synaptic markers in the *Quadratus lumborum* (QL), a disease-relevant axial muscle that is severely denervated in SMA mice[20,65]. AAV-GFP treated mice displayed 50% denervation in the QL muscle at P9, which was rescued in the SMN-WT treated SMA group, but only partially corrected in the AAV-SMN-2VA injected animals (Fig. 7b and d). Taken together, these results in SMA mice suggest that SIM-inactivated SMN is still functional and able to prevent the downstream cascade of events that lead to motor neuron death, but only partially those that cause NMJ denervation.

**SIM-inactivated SMN fails to rescue sensory-motor circuit dysfunction in SMA mice.** It has been shown that SMA motor neurons exhibit a reduction in their proprioceptive reflexes along

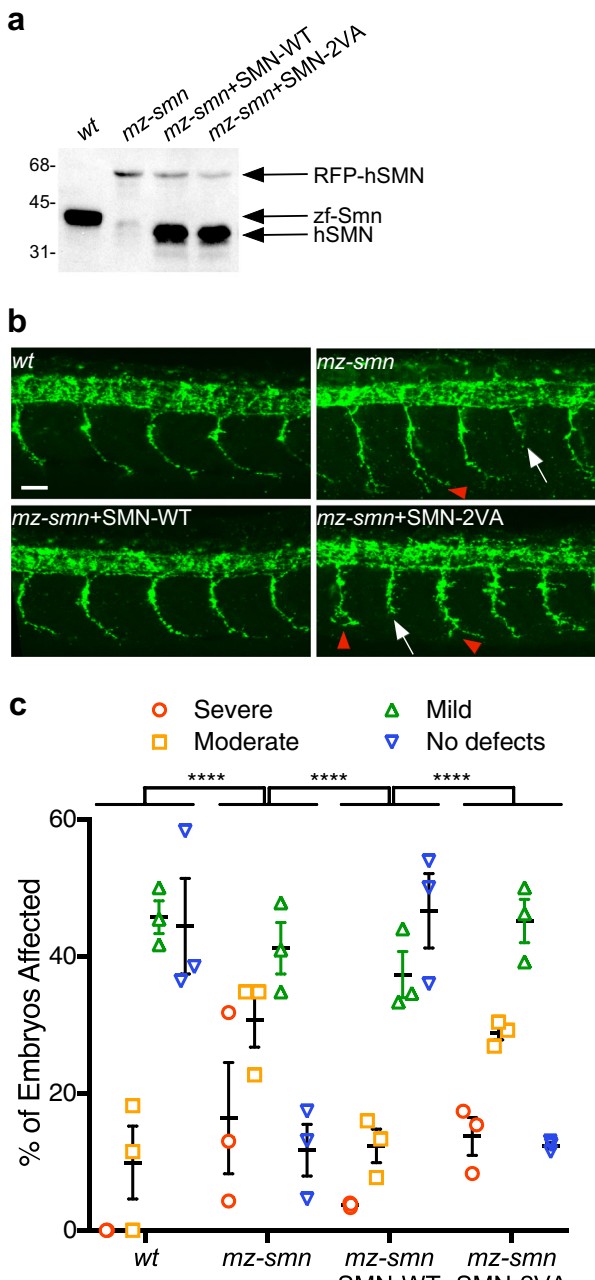

**Fig. 5 SIM-less SMN fails to rescue motor axon deficits in a zebrafish model of SMA. a** 250 pg of WT or 2 VA human SMN RNA was injected into one-cell stage *mz-smn* mutants zebrafish embryos and western blot analysis showed that both WT and mutant SMN protein are expressed at comparable levels. **b** Representative images of motor neuron phenotype for each group. Scale bar 50 µm. Truncated motor axons (white arrows) and branched motor axons (red arrowheads) occur, in relation to SMN levels reduction, with increasing degrees of severity determining the classification in one phenotypic class. **c** Animals were scored as having severe, moderate, mild or no defects in their motor axons at 28 h post fertilization (hpf). Data represent means and SEM. Three independent experiments were performed with 14–21 animals per experiment. Statistics were performed with two tailed Mann–Whitney non-parametric rank test (wt vs mz-smnp $p < 0.0001$; wt *vs* mz-smn + WT RNA $p = 0.865$; wt *vs* mz-smn + 2 VA SMN RNA $p < 0.0001$; mz-smn vs mz-smn + WT SMN RNA $p < 0.0001$; mz-smn vs mz-smn + 2 VA SMN RNA $p = 0.624$; mz-smn + WT SMN RNA vs mz-smn + 2 VA SMN RNA $p < 0.0001$).

with a loss of proprioceptive synapses on their soma and dendrites[61]. Interestingly, the loss of sensory synapses precedes and is independent from motor neuron loss in SMA mouse models[22,61,65] and postmortem human SMA spinal cords[69]. Thus, we focused on connectivity and function of proprioceptive synapses onto motor neurons controlling disease-relevant axial muscles. We quantified the number of proprioceptive synapses plotted against the transverse surface area of L1 motor neuron soma of P9 mice. Compared to controls, SMA mice showed a reduction of vesicular glutamate transporter 1 (VGluT1) proprioceptive synapses on the spinal motor neuron soma that was improved by WT-SMN treatment, but not in SMN-2VA treated mice (Fig. 8a and c).

Finally, we focused on the functionality of the sensory-motor circuits controlling disease-relevant axial muscles, which are severely disrupted in SMA mice. In agreement with the lack of morphological correction, the electrophysiological analysis revealed that SMN-2VA expression failed to improve central sensory-motor neurotransmission (spinal reflex amplitude, Fig. 8b and d). SMN-2VA treated animals did not display any rescue of proprioceptive inputs on motor neurons, suggesting that SMN-2VA does not exert any significantly beneficial effects on the central synapses impinging on motor neurons.

**SIM-inactivated SMN gene delivery fails to rescue select downstream RNA processing events disrupted by SMN deficiency in SMA mice.** Evidences are emerging for RNA processing events that have been mechanistically linked to specific SMN-regulated pathways and are responsible for select deficits in the sensory-motor circuit of SMA mice[9,21,22,70]. We, therefore, sought to investigate which of these RNA processing events are corrected by SMN-2VA expression as a way to link the loss of function of SIM-inactivated SMN to the disease phenotype. Previous work has demonstrated that SMN controls *Stas* expression through its role on U12 spliceosome biogenesis[21]. Interestingly, while *Stas* expression is corrected by SMN-WT, SIM-inactivated SMN is unable to rescue *Stas* aberrant splicing (Fig. 9a). This result links aberrant *Stas* processing by SMN-2VA with the failed correction of VGluT1 inputs on motor neuron soma of SMA mice, a finding supported by previous evidence that *Stas* contributes to the loss of sensory synapses in a mouse model of SMA[22].

We also monitored the effect of SMN-2VA expression on the U7-dependent 3′ end processing of histone mRNAs[8] and the p53-dependent upregulation of *Cdkn1a*[20,70]. While SMN-2VA expression corrects *Cdkn1a* upregulation to a level comparable to SMN-WT expression, histone *H1c* 3′ end processing is marginally improved by SMN-2VA expression (Fig. 9b and c). These results are consistent with the correction of motor neuron death by SMN-2VA expression and the role of p53 activation on neurodegeneration in SMA[20,70]. Taken together, these results indicate that loss of the SIM domain in SMN results in a hypofunctional SMN complex with limited capacity to rescue select aspects of SMA pathology (Fig. 9d).

## Discussion
Since the initial discovery of SMN as the SMA causing gene[71], research efforts have primarily focused on identifying SMN cellular functions and their contribution to the disease[3,4,16,72]. In contrast, little is known about the signaling pathways regulating SMN function. Here, we propose that sumoylation mechanistically links SMN-dependent downstream events with defined SMA phenotypes in cellular and animal models of the disease.

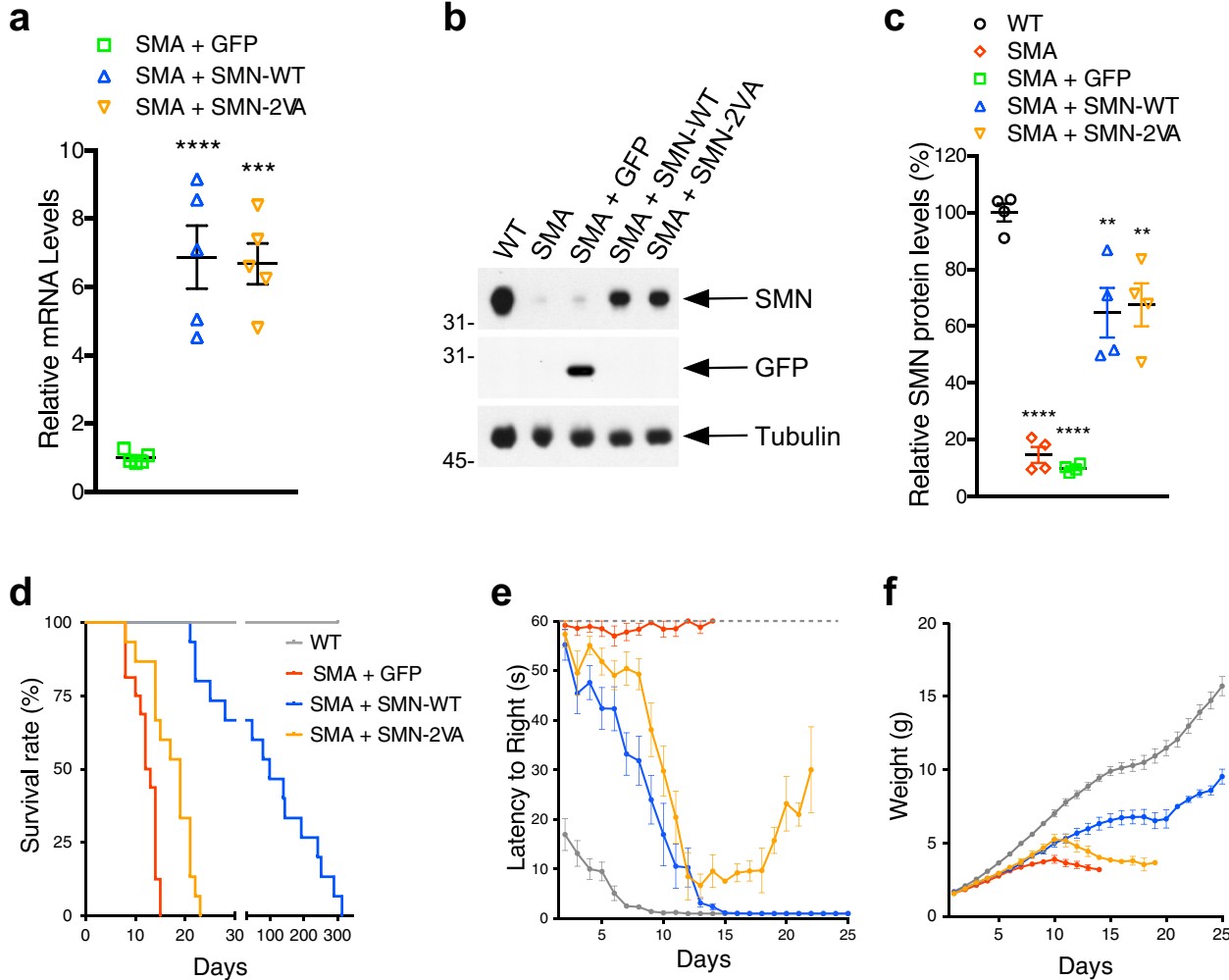

**Fig. 6 SIM-less SMN gene delivery marginally improves survival and motor function of SMA mice. a** RT-qPCR analysis of full-length human SMN mRNA in the spinal cord of SMA mice injected with AAV9-GFP, AAV9-SMN-WT or AAV9-SMN-2VA. Data represent mean and SEM ($n = 5$ animals). Statistical significance was determined by one-way ANOVA with Tukey's post hoc test (adjusted $P$ values: SMA + GFP vs. SMA + SMN-WT $p < 0.0001$; SMA + GFP vs. SMA + SMN-2VA $p = 0.0001$; SMA + SMN-WT vs. SMA + SMN-2VA $p = 0.9742$. **b** Western blot analysis of GFP and SMN WT or 2 VA protein levels in the spinal cord of AAV9-injected SMA mice and uninjected WT controls at P9. Tubulin is used as loading controls. **c** Quantification of four independent Western blot experiments. Data represent mean and SEM ($n = 4$). Statistical significance was determined by one-way ANOVA with Tukey's post hoc test (adjusted $P$ values: WT vs. SMA $p < 0.0001$; WT vs. SMA + GFP $p < 0.0001$; WT vs. SMA + SMN-WT $p = 0.0032$; WT vs. SMA + SMN-2VA $p = 0.0063$; SMA vs. SMA + GFP $p = 0.9723$; SMA vs. SMA + SMN-WT $p < 0.0001$; SMA vs. SMA + SMN-2VA $p < 0.0001$; SMA + GFP vs. SMA + SMN-WT $p < 0.0001$; SMA + GFP vs. SMA + SMN-2VA $p < 0.0001$; SMA + SMN-WT vs. SMA + SMN-2VA $p = 0.9962$). **d**–**f** Kaplan–Meyer analysis of survival (**d**; WT $n = 15$; SMA + GFP $n = 16$; SMA + SMN-WT $n = 15$, SMA + SMN-2VA n = 15), righting time (e; WT n = 15; SMA + GFP n = 13; SMA + SMN-WT $n = 17$, SMA + SMN-2VA $n = 18$), and weight gain (**f**; WT $n = 15$; SMA + GFP $n = 18$; SMA + SMN-WT $n = 17$, SMA + SMN-2VA $n = 17$). Data represent mean and SEM.

Our study shows that, in addition to being sumoylated, SMN can interact with SUMO through its SIM. SIM-inactivated SMN fails to assemble in a high molecular weight complex and has decreased activity in snRNP assembly. We also found that the SIM-inactivated mutant of SMN fails to localize in gems and Cajal bodies (CBs). Thus, a network of SUMO-SIM interactions likely contributes to the spatial organization of the SMN complex. In support of this view, we found that in addition to SMN, GEMIN3 and GEMIN5 are also sumoylated, and that their sumoylation is required for interaction with SMN. Accordingly, global inhibition of sumoylation through UBC9 knockdown in cells results in mislocalization of the SMN complex and impaired interaction of SMN with SUMO modifiable components of the complex. Remarkably, expression of SIM-inactivated SMN fails to rescue select phenotypes in two animal models of SMA, suggesting that loss of SUMO-SIM interactions leads to a

hypofunctional SMN complex. Taken together, these findings reveal a direct contribution of sumoylation to the integrity and function of the SMN complex as well as the biogenesis of Gems and CBs, adding these nuclear bodies to the short list of membrane-less organelles regulated by a network of SUMO-SIM interactions[43,73].

A previous study reported the presence of a SIM-like motif within the Tudor domain of SMN and that the loss of this motif impairs SMN's interaction with the essential components of snRNPs and CBs[26]. However, a clear demonstration that SMN's SIM is able to bind SUMO was lacking. Using recombinant proteins, here we showed that SMN contains a functional SIM that is able to interact with SUMO-2 and to a much lesser extent with SUMO-1. This finding adds SMN to the growing list of SUMO-2-binding proteins that are also sumoylated[50], supporting the idea that SUMO-SIM interaction stabilizes protein complexes

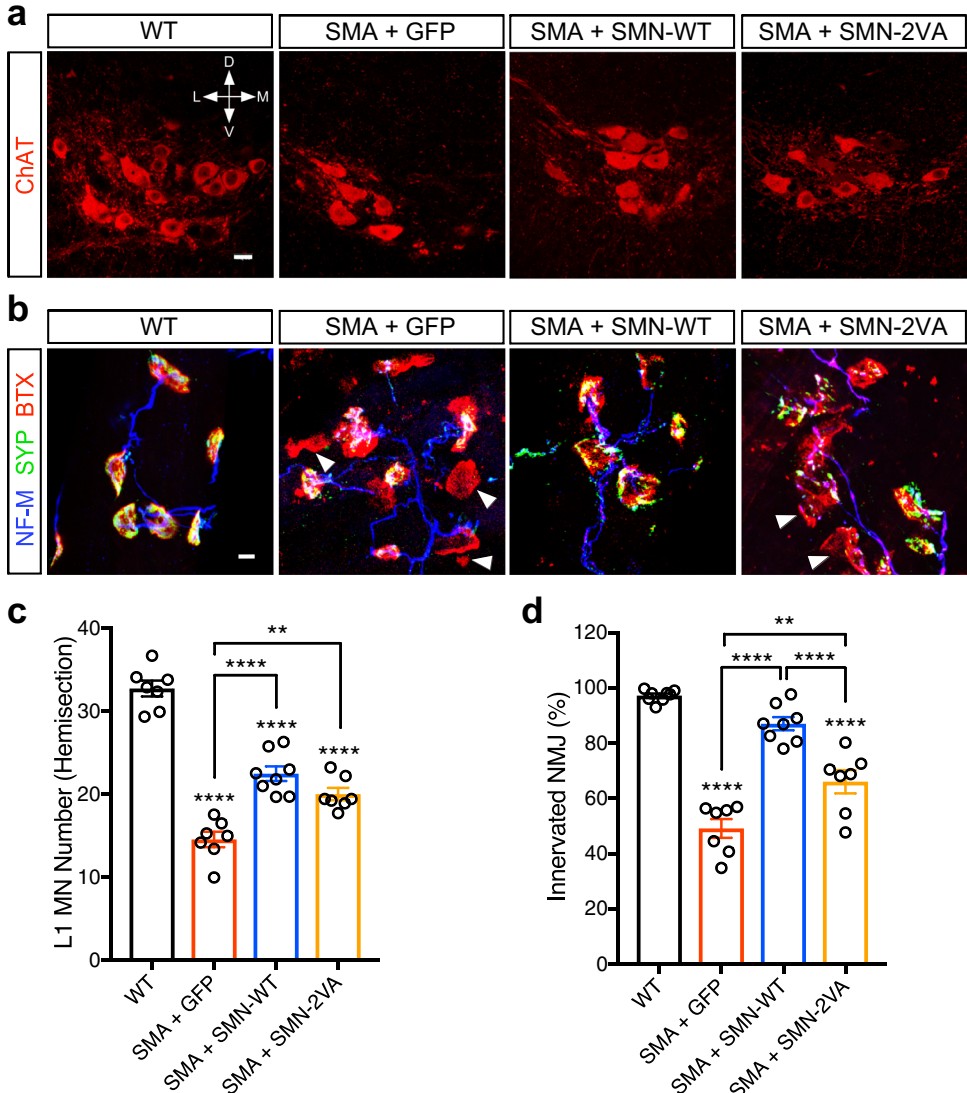

**Fig. 7 SIM-less SMN partially rescues motor neuron loss and NMJ denervation in SMA mice. a** ChAT immunostaining of L1 spinal cords from WT mice or SMA mice injected with AAV9-GFP, AAV9-SMN-WT, or AAV9-SMN-2VA at P9. Scale bar = 20 μm. **b** NMJ staining with bungarotoxin (BTX, red), synaptophysin (SYP, green), and neurofilament (NF-M, blue) of *quadratus lumborum* (QL) muscles from the same groups as in (a) at P9. SMN deficiency determines alterations at the NMJ level, affecting their proper innervation (denervated endplates appear as isolated red structures –white arrowheads– without colocalization between bungarotoxin and synaptophysin). Scale bar = 10 μm. **c** Total number of L1 motor neurons in the same groups as in (**a**) at P9. Data represent mean and SEM (WT $n = 7$ animals; SMA + GFP $n = 7$ animals; SMA + SMN-WT $n = 8$ animals; SMA + SMN-2VA $n = 7$ animals). Statistical significance was determined by one-way ANOVA with Tukey's post hoc test (adjusted *P* values: WT *vs.* SMA + GFP $p < 0.0001$; WT vs. SMA + SMN-WT $p < 0.0001$; WT vs. SMA + SMN-2VA $p < 0.0001$; SMA + GFP vs. SMA + SMN-WT $p < 0.0001$; SMA + GFP vs. SMA + SMN-2VA $p = 0.0012$; SMA + SMN-WT vs. SMA + SMN-2VA $p = 0.2171$). **d** Percentage of fully innervated neuromuscular junctions (NMJs) in the QL muscle from the same groups as in (**b**) at P9. Data represent mean and SEM (WT $n = 9$ animals; SMA + GFP $n = 7$ animals; SMA + SMN-WT $n = 8$ animals; SMA + SMN-2VA $n = 7$ animals). Statistical significance was determined by one-way ANOVA with Tukey's post hoc test (adjusted *P* values: WT vs. SMA + GFP $p < 0.0001$; WT vs. SMA + SMN-WT $p = 0.0878$; WT vs. SMA + SMN-2VA $p < 0.0001$; SMA + GFP vs. SMA + SMN-WT $p < 0.0001$; SMA + GFP vs. SMA + SMN-2VA $p = 0.0018$; SMA + SMN-WT vs. SMA + SMN-2VA $p < 0.0001$). Significant differences with WT have been reported immediately atop of each bar plot, while comparisons between the other experimental conditions have been graphed with connecting lines.

in a feedforward mechanism[74]. In line with this proposed mechanism is our observation that GEMIN3 and GEMIN5 are modified by SUMO-2 and can form poly-sumoylated chains. Poly-sumoylation may determine the fate of the SMN complex by enabling the participation of its components in different sub-complexes depending on their sumoylation levels and SUMO chain extension state[50,75]. In support of this view are our findings that both global sumoylation inhibition by UBC9 silencing and loss of SMN's SIM impair the integrity and subcellular distribution of the entire complex, although to a different degree of

impairment. One of the reasons for the differential effect on SMN ability to form macromolecular complexes may come from an incomplete inhibition of sumoylation by UBC9 silencing with some residual sumoylation that might render the phenotype less severe when compared to SIM-inactivated SMN in which all SUMO interactions are lost. Consistent with this interpretation, our data showed some residual binding of SMN to GEMIN5 and GEMIN3 upon UBC9 knockdown.

Global inhibition of sumoylation and SMN's SIM loss both result in an impaired localization of SMN in Gems. However,

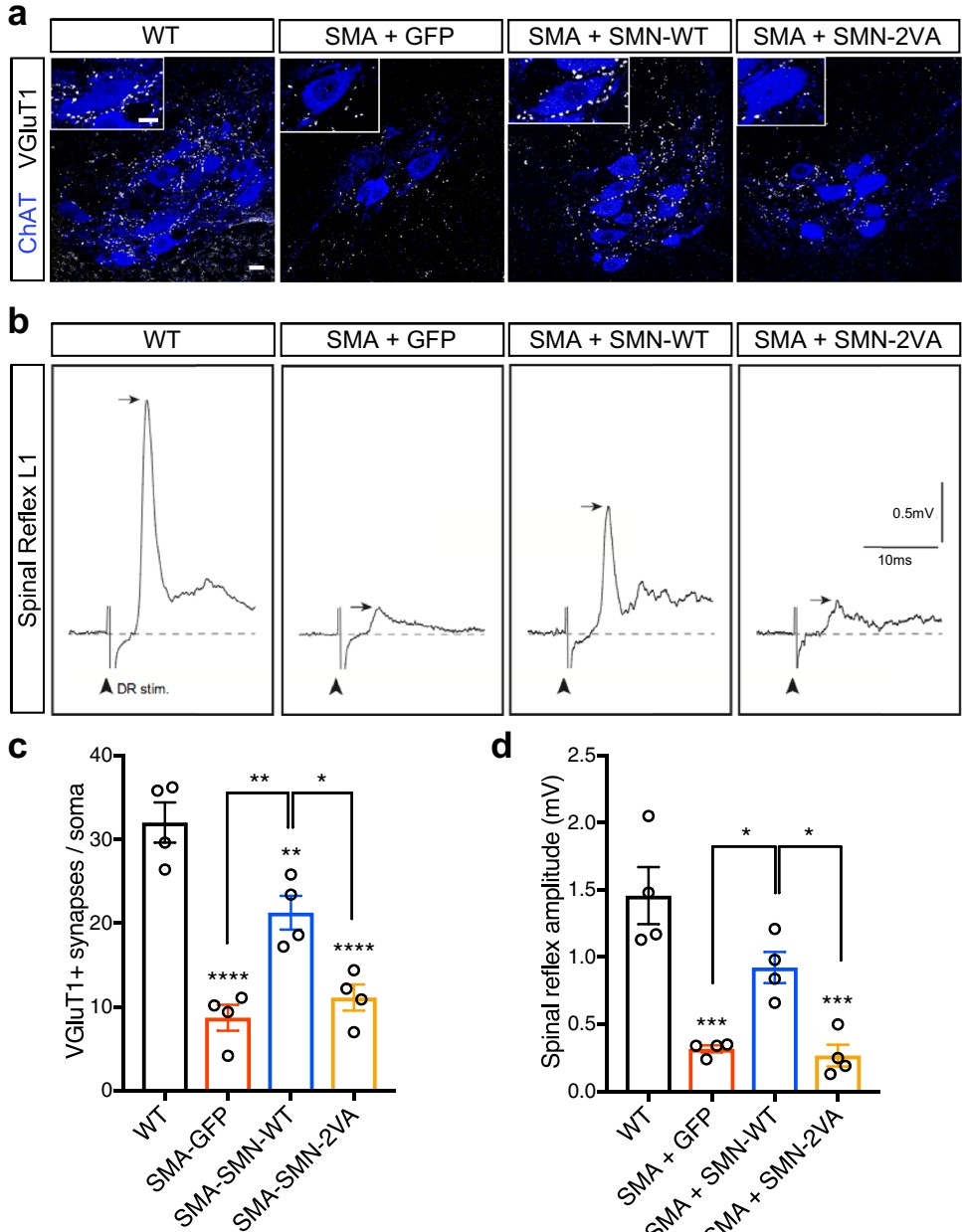

**Fig. 8 SIM-less SMN fails to rescue sensory-motor circuit dysfunctions in SMA mice. a** Immunostaining of VGluT1 + synapses (white dots) and ChAT + MNs (blue) in L1 spinal cord sections from WT mice or SMA mice injected with AAV9-GFP, AAV9-SMN-WT or AAV9-SMN-2VA at P9. Scale bar = 20 μm; scale bar of the insert = 10 μm. **b** Representative traces of extracellular recordings from L1 ventral root following L1 dorsal root stimulation from WT mice or SMA mice injected with AAV9-GFP, AAV9-SMN-WT or AAV9-SMN-2VA at P9. Arrows indicate the maximum amplitude of the monosynaptic reflex. Arrowheads indicate the stimulus artifact. Scale bars = 0.5 mV and 10 ms. **c** Number of VGluT1 + synaptic boutons (white dots) on the somata of L1 MNs (blue) from the same groups as in (**a**) at P9. Data represent mean and SEM from WT, SMA + AAV9-GFP, SMA + AAV9-SMN-WT, and SMA + AAV9-SMN-2VA (average of boutons counted for at least 15 cells from 4 animals for each group, $n = 4$ animals). Statistical significance was determined by one-way ANOVA with Tukey's post hoc test (adjusted $P$ values: WT vs. SMA + GFP $p < 0.0001$; WT vs. SMA + SMN-WT $p = 0.0085$; WT vs. SMA + SMN-2VA $p < 0.0001$; SMA + GFP vs. SMA + SMN-WT $p = 0.003$; SMA + GFP vs. SMA + SMN-2VA $p = 0.8208$; SMA + SMN-WT vs. SMA + SMN-2VA $p = 0.0132$). **d** Quantification of spinal reflex amplitudes recorded from the same groups as in (**b**) at P9. Data represent mean and SEM ($n = 4$ animals). Statistical significance was determined by one-way ANOVA with Tukey's post hoc test (adjusted $P$ values: WT vs. SMA + GFP $p = 0.0002$; WT vs. SMA + SMN-WT $p = 0.0633$; WT vs. SMA + SMN-2VA $p = 0.0002$; SMA + GFP vs. SMA + SMN-WT $p = 0.0254$; SMA + GFP vs. SMA + SMN-2VA $p = 0.9922$; SMA + SMN-WT vs. SMA + SMN-2VA $p = 0.0157$). Significant differences with WT have been reported immediately atop of each bar plot, while comparisons between the other experimental conditions have been graphed with connecting lines.

aberrant localization of the SMN complex within cytoplasmic foci is only observed upon global inhibition of sumoylation. This discrepancy is likely the result of sumoylation's well-established role in nucleocytoplasmic trafficking both through its effects on the properties of cargo proteins and also by regulating the

functions of components of the nuclear transport machinery[27]. Consistent with this possibility, we found that nucleocytoplasmic distribution of SMN is altered by UBC9 silencing, while SMN nuclear import is not affected by the loss of SUMO interaction, as reported in Tapia et al.[26]. Collectively, these results suggest that

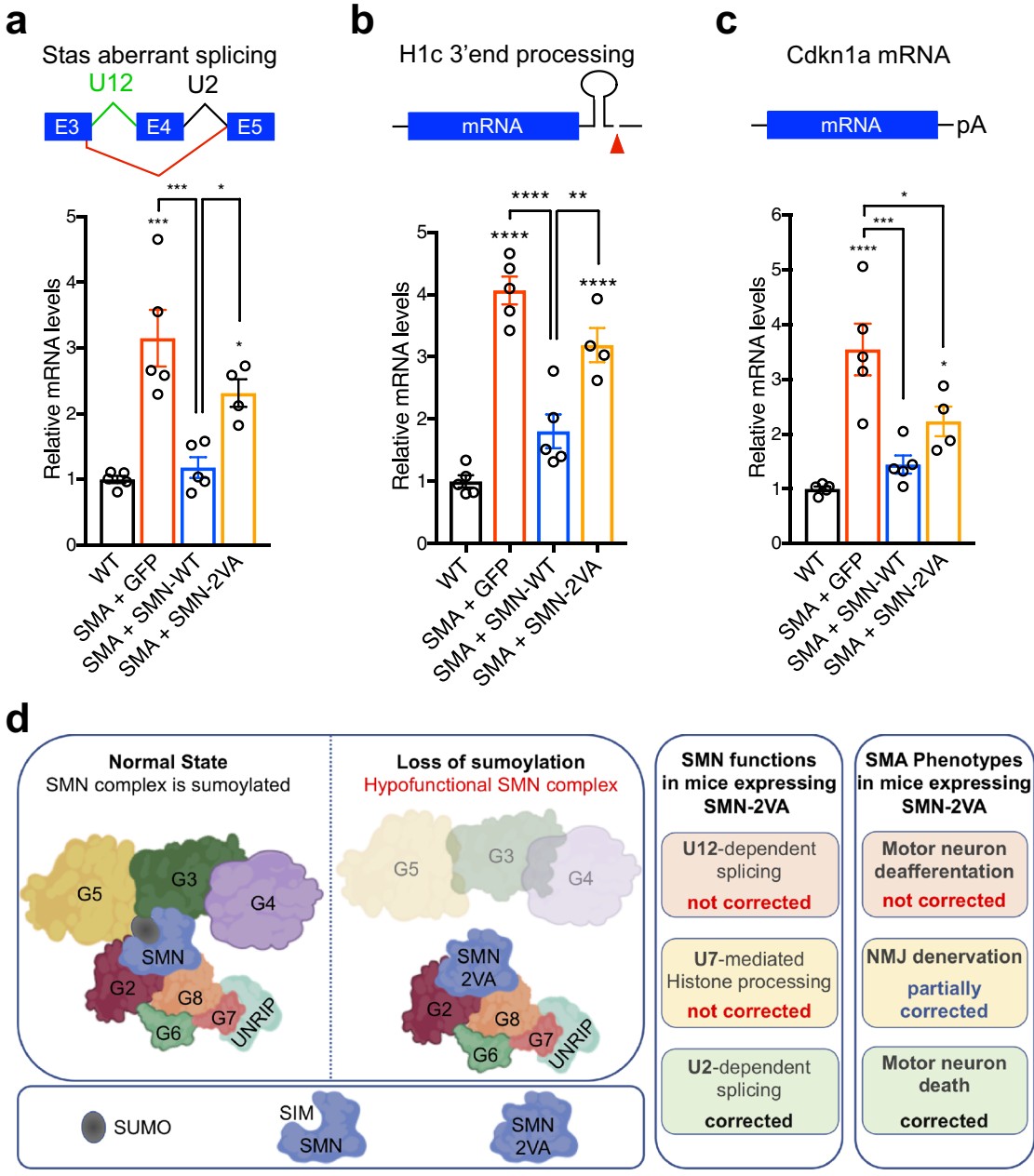

**Fig. 9 SIM-less SMN fails to rescue select downstream RNA processing events disrupted by SMN deficiency in SMA mice. a** RT-qPCR analysis of aberrantly spliced *Stasimon* (*Stas*) mRNA (indicated by the red lines between exon 3 and exon 5) in the spinal cord from WT mice or SMA mice injected with AAV9-GFP, AAV9-SMN-WT, or AAV9-SMN-2VA at P9. Data represent mean and SEM (WT $n = 5$ animals; SMA + GFP $n = 5$ animals; SMA + SMN-WT $n = 5$ animals; SMA + SMN-2VA $n = 4$ animals). Statistical significance was determined by one-way ANOVA with Tukey's post hoc test (adjusted *P* values: WT vs. SMA + GFP $p = 0.0001$; WT vs. SMA + SMN-WT $p = 0.9561$; WT vs. SMA + SMN-2VA $p = 0.0162$; SMA + GFP vs. SMA + SMN-WT $p = 0.0003$; SMA + GFP vs. SMA + SMN-2VA $p = 0.1682$; SMA + SMN-WT vs. SMA + SMN-2VA $p = 0.0408$). **b** RT-qPCR analysis of total 3′ end-extended histone H1c mRNA. Data represent mean and SEM (WT $n = 5$ animals; SMA + GFP $n = 5$ animals; SMA + SMN-WT $n = 5$ animals; SMA + SMN-2VA $n = 4$ animals). Statistical significance was determined by one-way ANOVA with Tukey's post hoc test (adjusted *P* values: WT vs. SMA + GFP $p < 0.0001$; WT vs. SMA + SMN-WT $p = 0.0827$; WT vs. SMA + SMN-2VA $p < 0.0001$; SMA + GFP vs. SMA + SMN-WT $p < 0.0001$; SMA + GFP vs. SMA + SMN-2VA $p = 0.0725$; SMA + SMN-WT vs. SMA + SMN-2VA $p = 0.0036$). **c** Cdkn1a mRNA in the spinal cord at P9 from the same treatment groups as in (**a**). Data represent mean and SEM (WT $n = 5$ animals; SMA + GFP $n = 5$ animals; SMA + SMN-WT $n = 5$ animals; SMA + SMN-2VA $n = 4$ animals). Statistical significance was determined by one-way ANOVA with Tukey's post hoc test (adjusted *P* values: WT vs. SMA + GFP $p < 0.0001$; WT vs. SMA + SMN-WT $p = 0.6815$; WT vs. SMA + SMN-2VA $p = 0.0464$; SMA + GFP vs. SMA + SMN-WT $p = 0.0005$; SMA + GFP vs. SMA + SMN-2VA $p = 0.0319$; SMA + SMN-WT vs. SMA + SMN-2VA $p = 0.2841$). Significant differences with WT have been reported immediately atop of each bar plot, while comparisons between the other experimental conditions have been graphed with connecting lines. **d** Model of the consequences of loss of SMN-SUMO interactions on the assembly and function of the SMN complex.

the impaired function and mislocalization of SIM-inactivated SMN are the results of inefficient interaction between SMN and the sumoylatable components of the complex, rather than a change in protein conformation in the SIM-inactivated mutant. Accordingly, SMA-causing mutations in the Tudor domain, such as SMN-E134K, do not impair SMN's interaction with GEMIN3 and GEMIN5, despite the fact that E134K mutation induces structural instability of the Tudor domain and disrupts SMN nuclear transport[55,56].

SMN deficiency elicits motor axon defects in zebrafish embryos that can be prevented by the expression of human SMN-WT[76,77]. Therefore, we investigated the activity of SIM-inactivated SMN using a maternal-zygotic SMN mutant of zebrafish[60]. We show here that sumoylation is required for normal motor axon outgrowth during zebrafish development as SIM-inactivated SMN expression is not able to correct the axonal defects in motor neurons of *smn* deficient embryos. These results further implicate SUMO interactions of SMN in motor neuron phenotypes of a vertebrate model of SMA.

Motor neuron loss is the hallmark of SMA pathology[16,17,66]. Activation of the tumor suppressor p53 drives motor neuron death in the SMNΔ7 mouse model of SMA and increased levels of *Cyclin-dependent kinase inhibitor 1a* (*Cdkn1a*) can be used as a molecular readout of the activation of this pathway[20,78]. Induction of p53 is mediated by SMN-dependent dysregulation of *Mdm2* and *Mdm4* alternative splicing[70]. Moreover, degeneration of vulnerable SMA motor neurons requires the convergence of p53 stabilization with one or more additional events, which include phosphorylation of p53 in its N-terminal trans-activation domain[20]. Our finding that AAV9-mediated expression of SIM-less SMN improves motor function by preventing motor neuron degeneration is consistent with a role for p53 activation in the demise of motor neurons as determined by the lack of *Cdkn1a* upregulation in the spinal cord of SMA mice expressing SMN-2VA.

One of the earliest manifestations of the disease in mouse models of SMA[61] is the sensory-motor dysfunction caused by the effects of SMN deficiency within proprioceptive neurons[65]. The resulting reduction of presynaptic glutamate release induces membrane hyperexcitability and reduces the firing ability of motor neurons, thereby contributing to impaired muscle contraction in SMA mice[65]. We previously identified *Stasimon* (also known as *Tmem41b* or *Stas*) as an evolutionarily conserved, U12 intron-containing gene regulated by SMN that is required for proper synaptic transmission in the motor circuit of *Drosophila* larvae and for motor axon outgrowth in zebrafish[21]. Furthermore, we and others have found that SMN deficiency disrupts U12 splicing[12,13,21,68], and both misprocessing and reduced expression of *Stas* mRNA was found in disease-relevant motor circuit neurons of SMA mice[21]. Moreover, *Stas* gene delivery rescues the loss of proprioceptive synapses on motor neurons of SMN mice[22]. Here, we found that AAV-mediated expression of SIM-less SMN neither rescues the loss of proprioceptive synapses on motor neurons nor improves sensory-motor synaptic transmission. Importantly, our findings that *Stas* expression and processing are not corrected by SIM-less SMN expression are consistent with its contribution to defective synaptic connectivity and function in the SMA sensory-motor circuit[21,22]. However, in addition to rescue sensory-motor connectivity, AAV-mediated gene delivery of *Stas* suppresses motor neuron degeneration by preventing the signaling cascade that feeds into the p53-mediated pathway of death of SMA motor neurons through p38 MAPK activation[22]. Hence, our results that SIM-less SMN preserves motor neurons even without restoring *Stas* expression suggest that it acts on the Mdm2/4 axis leading to p53 induction and support the view that death of SMA motor neurons requires both stabilization and activation of the p53 pathway[20,70].

Our findings that loss of SUMO-SMN interaction affects some, but not all of the disease-related deficits in SMA mice support the view that a combination of defects involving SMN-dependent RNA pathways contribute to SMA pathology[17,79]. This differential effect on the various pathological features of SMA can depend on two non-mutually exclusive hypotheses. On one hand, it is possible that sumoylation is more prominently required for specific pathways controlled by the SMN complex, such as the U12-dependent splicing, which would cause the loss of sensory inputs on motor neurons because of the perturbation of the expression of *Stas*. On the other hand, is possible that SIM-inactivated SMN is a partial loss of function and its expression may cause more limited rescue of SMN-dependent RNA pathways that have a higher sensitivity to SMN reduction. In this scenario, RNA-mediated pathways with the highest dependence on SMN activity may be least affected by moderate increases in SMN function. Our results that *Stas* expression is not corrected by SIM-inactivated SMN, are consistent with U12-dependent splicing being highly susceptible to SMN deficiency in SMA mice[12–14,80].

In conclusion, here we show that sumoylation of Gemins and the SUMO-binding properties of SMN work in concert to underlie the integrity and function of the SMN complex in snRNP biogenesis. Moreover, sumoylation participates in the SMN activities that are relevant for SMA. Our findings provide novel insights into disease mechanisms and also highlight the role of PTMs in determining the interaction and functioning of multiprotein complexes that are essential for life. Based on these observations, a deeper understanding of the role of sumoylation on SMN biology will be important to fully dissect the multifaceted functions of SMN and their link to SMA pathology and ultimately broaden the knowledge necessary for the development of increasingly effective therapies for SMA.

## Methods

**Antibodies**. The antibodies used in this work are listed in Supplementary Table 3.

**Primers**. All primers used in this work are listed in Supplementary Table 4.

**DNA constructs**. For lentiviral constructs, with the exception of the commercially available pLenti6/TR (Invitrogen), all other constructs were generated by standard cloning techniques using the pRRLSIN.cPPT.PGK-GFP.WPRE vector (Addgene pasmid 12252) as a backbone. The pLenti6/TR construct constitutively expresses the tetracycline-dependent repressor (TetR) protein under the control of the CMV promoter as well as the blasticidin resistance gene from the SV40 promoter. The pLenti.pur/SmnRNAi construct expresses an shRNA targeting mouse UBC9 mRNA under the control of a tetracycline-regulated H1TO promoter as well as the puromycin resistance gene from the phosphoglycerate kinase (PGK) promoter. DNA fragments corresponding to the ORFs of GFP and human SMN WT or 2 VA were cloned downstream of the GUSB promoter of vectors harboring AAV2 inverted terminal repeats (ITRs) for the production of self-complementary AAV9. To generate SMN and GEMIN5 mutants, PCR site-directed mutagenesis was performed on the respective wild-type plasmid with the oligos reported in Supplementary Table 4. To generate GEMIN5-2KR, site-directed mutagenesis was performed with oligos G5-K828R using GEMIN5-K754R cDNA as a template. The desired mutations were confirmed by DNA sequencing. GST-fused proteins were generated by PCR using SMN WT or 2 VA as template and cloned into the prokaryotic expression vector pGEX4T1 (Cytiva). All constructs were verified by DNA sequencing.

**Lentiviral production**. Helper plasmids pLP1, pLP2, pLP.VSV.G and backbone plasmid were co-transfected in 293T cells using calcium phosphate (CalPhos Mammalian Transfection Kit, Clontech). After 2 days, lentivirus in supernatants were harvested and concentrated by centrifugation. The viral titer was determined with the Lenti-XTM qRT-PCR Titration Kit (Clontech). Four hours post-plating, HeLa and NIH3T3 cells were infected with serial dilutions of viral particles.

**AAV9 production**. All AAV preps were produced by Virovek (Hayward, CA). DNA for the production of scAAV9 vectors was purified using endotoxin-free Mega preparation kit (Qiagen) according to the manufacturer's instructions, then used to produce AAV vectors through Virovek's proprietary Baculovirus system.

Virions were purified through CsCl density ultra-centrifugation method, buffer-exchanged, and sterilized. The resulting vectors were concentrated to final titers of $\sim 2 \times 10^{13}$ genome copies per milliliter using Amicon Ultracel centrifugal filter devices with a 30,000 nominal molecular weight limit (Millipore). The titers were determined by qPCR assay, and the purity by an SDS-PAGE gel, followed by SimplyBlue-Staining.

**Cell culture**. NIH3T3, HeLa and HEK-293T cells were grown in Dulbecco modified Eagle medium (DMEM) with high glucose (Gibco) containing 10% fetal bovine serum (HyClone), 2 mM glutamine (Gibco), and 0.1 mg/mL gentamicin (Gibco). For SMN knockdown studies, RNA interference (RNAi) was induced where needed by addition to the growth medium of doxycycline (Fisher) at a final concentration of 100 ng/mL, and cells were collected after 5 days unless otherwise indicated as previously described[21].

**RNA analysis**. Total RNA from mouse NIH3T3 fibroblasts and mouse spinal cord was isolated using TRIzol reagent (Invitrogen) and following RNase-free DNaseI (Ambion) digestion. RNA concentrations were determined using a NanoDrop 1000 (Fisher Scientific). cDNA was then generated with Advantage RT-for-PCR kit (Clontech) using a mixture of oligo-dT primers, random hexamers, and 2.5 µg of total RNA, following manufacturer's instructions. Primers sequences used in this study are reported in Supplementary Table 4. Each measurement was performed in triplicates in a standard 3-step qPCR reaction with a Mastercycler ep Realplex[4] (Eppendorf) PCR system and Power SYBR® Green PCR Master Mix (ABI).

**Protein analysis**. For Western blot analysis, the protein was extracted from cells and tissues and analyzed according to established procedures[21]. Cells were lysed in SDS sample buffer (2% SDS, 10% glycerol, 5% β-mercaptoethanol, 60 mM Tris-HCl, pH 6.8, bromophenol blue) unless otherwise indicated, and passed through a 27-gauge needle five times. Protein extracts were quantified using the RC DC protein assay (Bio-Rad). Samples were run on a 12% SDS-PAGE gel and transferred onto a Trans-Blot transfer medium nitrocellulose membrane (Bio-Rad) using a TE77x semidry transfer unit (Hoefer) using 1x Tris-glycine buffer (Bio-Rad) containing 20% methanol. Membranes were stained with 0.1% (wt/vol) Ponceau S and 5% acetic acid, destained in distilled H2O, and blocked for 1 h at room temperature with 5% nonfat dry milk (LabScientific) in PBS containing 0.1% Tween 20 (Acros). Incubation with primary antibody was performed in PBS containing 0.1% Tween 20 for 1 h at room temperature. Membranes were washed 3 times for 10 m with PBS containing 0.1% Tween 20 at room temperature. Incubation with secondary antibodies conjugated to horseradish peroxidase was performed in PBS containing 0.1% Tween 20 at room temperature. Membranes were washed three times for 10 m with PBS containing 0.1% Tween 20 at room temperature. Chemiluminescence was carried out using a SuperSignal West Pico chemiluminescent substrate (Thermo Scientific) according to the manufacturer's instructions. Signal was detected by autoradiography using Full Speed Blue sensitive medical X-ray film (Ewen Parker X-Ray Corporation). Unprocessed scans of the most important blots have been included in the Source Data file.

To determine SMN protein stability, NIH3T3-SMN/Smn$_{RNAi}$ cells (wild-type and SMN-2VA) were grown in the presence or absence of Doxycycline. They were plated with a concentration of 100,000 cells in 10 cm dishes. Cells were treated with Cycloheximide (100 µg/mL) at different time points. After Cycloheximide treatment cells were collected and analyzed by Western blot, using an anti-strep tag antibody (Qiagen).

Immunoprecipitation experiments were carried out from NIH3T3 total cell extracts using either antibodies bound to protein G-Sepharose (Sigma) or FLAG M2-agarose (Sigma) in RSB-100 buffer containing 0.1% Nonidet P-40 for 2 h at 4 °C. Three washes with RBS-100 0.02% NP40, three with RSB 500 0.02% NP-40 and one with RSB-100 0.1% NP40 were performed. For FLAG immunoprecipitates elution, FLAG peptides in RSB-100 0.1% NP40 were added to the beads (for 30 min at 4 °C). Samples were then boiling in SDS sample buffer and analyzed by SDS/PAGE on 12% polyacrylamide gel followed by Silver Staining or western blotting.

For sucrose gradient centrifugation experiments, cell protein extract was prepared as described above and an equal amount of protein extracts were fractionated on 10 ml of 10–30% sucrose gradients in RSB-100 buffer by centrifugation for 4 h at 38,000 rpm in an SW 41 rotor at 4 °C. Gradient fractions were collected (600µl for each fraction) and samples were diluted in sample buffer and then run on a 12% SDS-PAGE gels foe Western blot analysis as described above.

**Protein expression, purification, and in vitro pull-down assays**. Overnight starter cultures (50 ml) of BL21 (DE3) transformed with plasmid expressing GST or GST-fused SMN-Tudor domain, or His-fused SUMO1 or SUMO2 protein were individually inoculated 500 ml of Luria Broth (LB) culture medium with specific antibiotic and grown at 30 °C to a density of appropriately 0.6 optical density at 600 nm. After 1 mM isopropylthiogalactopyranoside (IPTG) induction at 30 °C for 4 h, the bacteria were collected and sonicated in lysis buffer (20 mM Tris-HCl pH 8.0, 100 mM NaCl, 0.5% NP40, 1 mM EDTA, 1 M DTT, 5% Sarkosyl and protease inhibitors). Recombinant proteins GST, GST-SMN-TD (WT or 2 VA) were purified by Glutathione Sepharose chromatography according to the manufacturer's

instruction (Cytiva). For pull-down assay, cell extracts were individually incubated with His-fusion proteins loaded on beads for 3-h at 4 °C in binding buffer (50 mM Tris-HCl pH 7.5, 100 mM NaCl, 10 µm ZnCl₂, 10% glycerol, freshly supplemented with 0.1 mM Dithiothreitol and protease inhibitors). After washing, bound proteins were eluted with SDS sample buffer and analyzed by gel electrophoresis followed by Coomassie staining or immunoblotting with specific antibodies.

**snRNP assembly assay**. U1 snRNA was transcribed in vitro from linearized template DNAs in the presence of [α-$^{32}$P]UTP (3000 Ci/mmol) and purified from denaturing polyacrylamide gels according to standard procedures. Extracts for snRNP assembly were prepared as described previously with minor modifications[6]. NIH3T3 cells were resuspended in ice-cold reconstitution buffer (20 mM Hepes-KOH, pH 7.9, 50 mM KCl, 5 mM MgCl2, 0.2 mM EDTA, 5% glycerol) containing 0.01% Nonidet P-40, passed five times through a 25-gauge needle, and then centrifuged at 10,000 rpm for 15 min at 4 °C. Supernatants were collected and used for snRNP assembly experiments. snRNP assembly reactions with cell extracts were carried out for 1 h at 30 °C in a volume of 20 µl of reconstitution buffer containing 0.01% Nonidet P-40, 10,000 cpm of radiolabeled U1 snRNA, 2.5 mM ATP, and 10 µM E. coli tRNA. Reactions were then immunoprecipitated for 2-h at 4 °C with anti-SmB (18F6) antibodies bound to protein G-Sepharose in RSB-500 buffer (500 mM NaCl, 10 mM Tris-HCl, pH 7.4, 2.5 mM MgCl2) containing 0.1% Nonidet P-40 and protease inhibitors. Following five washes with the same buffer, bound RNAs were recovered from immunoprecipitates by proteinase K treatment, phenol/chloroform extraction, and ethanol precipitation. RNAs were then analyzed by electrophoresis on denaturing polyacrylamide gels and autoradiography.

**Bioinformatics**. Human SMN predicted sumoylation target-sequences were identified with bioinformatics analysis using the SUMOplot software (http://www.abgent.com/sumoplot/), Joined Advanced Sumoylation Site, and Sim Analyser (JASSA, version 4 - http://www.jassa.fr), GPS-SUMO (http://sumosp.biocuckoo.org) according to the toll instructions. All the predicted SUMO target sequences for the target genes (SMN1, Gemin2, Gemin3, Gemin4, Gemin5, Gemin6, Gemin7, Gemin8, UNRIP) were annotated (Supplementary Table 1). For each target sequence the score reported by each bioinformatic tool was collected (reference score for each tool: SUMOplot: 0-1.0, JASSA: low vs high, GPS-SUMO: p-value). Sequences reported as "SUMO nonconsensus" with the GPS-SUMO tool were not considered, unless also reported in at least one of the other two tools. Published proteomics data relative to the sumoylation of the proteins of the SMN-complex were reviewed. We compared the predicated data of sumoylation of the SMN complex components (as reported in Supplementary Table 1) and the ones reported in the database http://proteomecentral.proteomexchange.org/cgi/GetDataset. Datasets were filtered using the searching term "sumoylation", which identified 26 additional works. Overlaps with bioinformatically predicted sites are detailed in Supplementary Table 2.

**In vitro sumoylation assay**. In vitro SUMO modification reactions were performed as indicated in Lee et al.[49]. Recombinant SUMO and the other enzymes that will be used in the following experiments were previously expressed in E.Coli and purified as previously described[49]. Briefly, cDNAs coding for SUMO-2 and GEMIN5 were transcribed and translated in rabbit reticulocyte lysate in the presence of [$^{35}$S]methionine according to the manufacturer's instructions (Promega). Assays were performed in the presence of high concentrations of E1 (220 nM) and E2 (600 nM) enzymes and contained 2 µl of translation product in a 10-µl reaction mixture containing 20 mM HEPES (pH 7.3), 110 mM potassium acetate, 2 mM magnesium acetate, 1 mM dithiothreitol (DTT), 10 µM recombinant SUMO-1, 1 mM ATP, 5 mM phosphocreatine (Sigma), 20 U of creatine phosphokinase (Sigma)/ml, and 0.6 U/ml of inorganic pyrophosphatase (Sigma)/ml. Reactions were separated by SDS-PAGE and analyzed by autoradiography. The process was repeated with individual and combined mutants for lysines which are predicted target locations for sumoylation.

**Zebrafish analyses**. Wild-type and maternal:zygotic (mz) smn mutants were used for the analyses (Hao et al 2013). mz-smn mutants are deficient of zebrafish smn but hold a transgene Tg (hsp70:RFPSMN) expressing low levels of human RFP-SMN. All animals were grown in the Ohio State University zebrafish facility under established protocols and OSU animal welfare guidelines. mRNA was generated from linearized DNA plasmids; wild-type human SMN, SMN-A124V/A125V in the pcDNA vector were linearized with SmaI. One–two-cell stage mz-smn embryos were treated with 250 pg of each mRNA separately. At 28 hpf, injected and control embryos were anesthetized using tricaine (250 µg/ml, Sigma A-5040) and fixed overnight at 4 °C in 4% formaldehyde. Motor axons responsible for the mid-trunk innervation on both sides were assessed as previously reported[81]. Western blot analyses were performed at 28 hpf using the mouse anti-SMN (1:500; MAN-SMA12) antibody, as described[59], with samples from embryos injected with mRNA at the one- to two-cell stage and uninjected wild-type embryos.

**SMA mice and experimental procedures**. All mice were handled according to the regulatory guidelines of the National Institutes of Health Guide on the Care and Use of Animals and approved by the Institutional Animal Care and Use Committee of Columbia University. Mice were housed in an animal facility controlled

for humidity (40–60%) and temperature (~18–23 °C) with a 12 h-12 h light-dark cycle with free access to food and water. The SMNΔ7 mouse line (Smn$^{+/-}$/ SMN2$^{+/+}$/SMNΔ7$^{+/+}$) used in this study to generate SMA mice was on a pure FVB background and was obtained from Jackson Mice (Jax stock no. 005025). All experiments included both males and females, and since gender-specific differences have not identified our data have been aggregated. Tail DNA PCR genotyping protocol has performed the primers listed in Supplementary Table 4 as described previously[65]. For intracerebroventricular (ICV) viral injection, P0 pups were anesthetized by isoflurane inhalation and 5 μl of AAV9-GUSB hSMN-WT, AAV9-GUSB hSMN-V2A, or AAV9-GUSB GFP virus were injected in the right lateral ventricle of the brain of SMAΔ7 mice and controls, using a modified Hamilton syringe. After 30 min of recovery, pups were placed in their breeder cage and monitored daily for weight and righting time. For AAV9 gene delivery, we delivered $1 \times 10^{11}$ genome copies of the indicated AAV9 vectors in a PBS solution containing a vital dye (Fast Green, Sigma). Mice were sacrificed, and tissue collection was performed in a dissection chamber under continuous oxygenation (95% O$_2$/5% CO$_2$) in the presence of cold (~12 °C) artificial cerebrospinal fluid (aCSF) containing: 128.35 mM NaCl, 4 mM KCl, 0.58 mM NaH2PO4, 21 mM NaHCO3, 30 mM D-glucose, 1.5 mM CaCl2, and 1 mM MgSO4 as previously reported[65].

**Animal survival and motor behavior**. Mice from all the experimental groups were weighted daily starting from postnatal day 1 (P1) and the life span was recorded (animals were sacrificed if a greater than a 20% of body weight was recorded from the previous day). Righting reflex was employed to analyze the motor behavior of the animals, as previously described[61,65]. Briefly, the animal was placed on its back and the time until all four limbs were placed on the ground was measured (cut-off 60 s due to ethical constrains imposed by IACUC guidelines; the procedure was repeated three times for each animal, allowing sufficient time for recovery between measurements).

**Immunohistochemistry and immunofluorescence analysis**. For spinal cord analysis, tissues were collected and fixed with paraformaldehyde (PFA) 4% overnight, then washed three times with PBS. The L1 spinal cord segment was identified and immersed in warm 2.5% agarose (LE Agarose GeneMate) gel and entirely sectioned at the VT1000 S vibratome (Leica) (at 70 μm thickness for each section). Free-floating spinal cord sections were blocked for 1 h at room temperature using a 24-well plate with 5% Donkey serum and 0.5% Triton X-100 in TBS. Primary antibodies (Supplementary Table 3) were incubated for 2 nights at 4° on an orbital shaker. The following day the sections were washed six times 20 m each with 0.5% Triton X-100 in TBS. Six washing steps of 10 min each were done prior to incubation with secondary antibodies for 3 h at room temperature in 5% Donkey serum 0.5% Triton X-100 in TBS. Another six washing steps were performed before sections were mounted in Fluoromont G (eBioscience). This procedure was applied also for DRG sections' staining. Isolated L1 DRGs were sectioned at the Vibratiome (60 μm each section) and immunostained following the protocol described above.

For NMJ analysis, isolated QL muscles were fixed with PFA for 15 m then washed with PBS and manually teased into small groups of single fibers. Muscle fibers were incubated with Bungarotoxin-Alexa 555 (Thermo Fisher) in PBS (1:250) overnight. The following day, fibers were washed with PBS and permeabilized with ice-cold methanol for 5 m. After washing, fibers were blocked with 10% donkey serum in 0.2% Triton-X-100 in PBS for 1 h at room temperature. Subsequently, fibers were incubated with primary antibodies (Supplementary Table 3) overnight at 4° in 10% donkey serum + 0.2% Triton-X100 in PBS. Finally, secondary antibodies were applied for 3 h at room temperature in 10% donkey serum + 0.2% Triton-X100 in PBS. After washing, fibers were mounted on coverslips with Fluoro-Gel with Tris Buffer (Electron Microscopy Sciences).

For immunofluorescence analysis, NIH3T3 or HeLa plated on coverslips were fixed in 4% PFA for 15 min and then permeabilized with 0.5% Triton X-100 in PBS for 10 min at 4 °C. After blocking for 30 m at room temperature in 3% BSA in PBS, cells were incubated with primary antibodies in 3% BSA in PBS overnight at 4 °C. Following three 5-min washes, cells were incubated with secondary antibody and DAPI in 3% BSA in PBS for 1 h, washed three times, and mounted using ProLong™ Gold Antifade Mountant (Thermo Fisher).

**Confocal imaging and analyses**. All images were acquired using a Leica SP5 confocal microscope and analyzed offline using the Leica LAS AF and Fiji software from z-stack images as described previously[61,65]. Motor neurons were counted from Z stack images (acquired at 1 μm intervals in the z-axis) collected for each section, which was scanned using a 40x objective. Only motor neurons (ChAT+) that contained a clearly visible nucleus (identified by being devoid of ChAT immunoreactivity) were considered in order to avoid double counting the same neuron from adjoining sections. Quantitative analysis of VGluT1 stained (VGluT1+) synaptic contacts on L1 spinal MNs at P9 were performed on z-stacks of optical sections acquired using a ×40 objective (acquired at 0.35 μm intervals in the z-axis) throughout the whole spinal cord section as to ensure the inclusion of the entire motor neuron soma of stained ChAT+ motor neurons. VGluT1+ synapses were counted off-line over each motor neuron soma using the Leica LASAF software. For the analysis of NMJ innervation, NMJ synapses were acquired with ×40 objective, and z-stack images were scanned at 1μm interval. NMJs were counted as innervated when the presynaptic neuronal terminal marker co-localized with the postsynaptic endplate. Care was taken to differentiate between fully denervated from either fully or partially innervated NMJ synapses.

**Electrophysiology**. Electrophysiological experiments were carried out as described previously[20,70]. For recordings of the monosynaptic reflex, the intact ex vivo spinal cord preparation was perfused continuously with oxygenated (95% O$_2$/5% CO$_2$) aCSF at ~10 mL/min. The dorsal root and ventral root of the L1 segment were placed into suction electrodes for stimulation and recording, respectively. The extracellular recorded potentials were acquired (DC, 3 kHz, Cyberamp, Molecular Devices) in response to a brief (0.2 ms) stimulation (A365, current stimulus isolator, World Precision Instruments) of the L1 dorsal root. The maximum responses were acquired following supramaximal stimulation intensity (ranging between 2 and 10 times the Threshold of stimulation). The stimulus Threshold was defined as the current at which a minimal evoked response was recorded in three out of five trials. Recordings were fed to an A/D interface (Digidata 1440A, Molecular Devices) and acquired with Clampex (version 10.2, Molecular Devices) at a sampling rate of 10 kHz. Data were analyzed offline using Clampfit (version 10.2, Molecular Devices). Measurements were taken from averaged traces of five trials elicited at 0.1 Hz. The temperature of the physiological solution ranged between 21 and 25 °C.

**Statistics and reproducibility**. Continuous variables were reported as percentages or means ± standard error of the mean (SEM), as appropriate. Differences between means have been analyzed using the 2-tailed Student's $t$ test. Multiple unpaired $t$ tests were performed with Welch correction (analyzing each raw individually without assuming consistent standard deviation) and corrected for multiple comparisons using the Holm-Sidak method. Differences among means were analyzed by one-way ANOVA. When the ANOVA showed significant differences, pair-wise comparisons between means were assessed using Tukey's post hoc testing. In all analyses, the null hypothesis was rejected at the 0.05 level and exact $P$ values were reported for each experiment in the figure legend. Significant differences with the control group have been reported immediately atop of each bar plot, while comparisons between the other experimental conditions have been graphed with connecting lines. Data sets violating normality and equality of variance have been analyzed with the two-tailed Mann–Whitney non-parametric rank test. All analyses were performed with GraphPad Prism version 7 (GraphPad Inc. La Jolla CA, USA).

All the experiments were repeated at least three times, with reproducible results within the variability of the biological experiments.

**Reporting summary**. Further information on research design is available in the Nature Research Reporting Summary linked to this article.

## Data availability
The authors declare that all the main data supporting the findings of this study are available within the article and its Supplementary Information files. Source data are provided with this paper.

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

## Acknowledgements

We are grateful to Dr. Livio Pellizzoni for the kind gift of the SMN (7F3), Gemin2 (14G1), SmB (18F6) antibodies, SMN-E134K cDNA, and for critical comments of the manuscript. We thank Dr. Michael Matunis for the kind gift of His-SUMO1, His-SUMO2, and the recombinant proteins for the in vitro sumoylation assay. We thank Benjamin Hoover and Manuel Alfredo Podestà for their critical reading of the manuscript. This work was supported by a grant from Cure SMA (F.L.), Project ALS (F.L.), and by NIH grants R01NS078375 and R01AA027079 (G.Z.M.) and R21NS101575 (F.L.). G.M.R., I.F. P.R., and S.C. received funding from the European Union's FP7-PEOPLE-2013-IRSES grant agreement No. 612578 and the Horizon 2020 research and innovation programme under the Marie Sklodowska-Curie grant agreement No 778003. This work is dedicated to the memory of Christine Beattie, our cherished colleague and beloved friend.

## Author contributions

F.L. designed and supervised the study. G.M.R., I.F., T.K., N.D., G.N., M.D.P.-S., P.R., and L.T.H. performed the experiments and analyzed the data. C.C.B., S.C., S.P., and G.Z.M. contributed to the design, analysis, revision, and interpretation of data. G.M.R., I.F., and F.L. wrote the paper with input from all authors.

## Competing interests

The authors declare no competing interests.
