## [Peer Review File · Nature Communications]

REVIEWER COMMENTS

Reviewer #1 (Remarks to the Author):

In the first part of the manuscript “Sumoylation regulates the assembly and activity of the SMN complex”, the authors show that mutating the SUMO-interacting motif (SIM) in SMN alters its sub-cellular distribution, SMN/Gemin complex assembly, and its function in snRNP biogenesis. The second part is more focused on animal models. The authors found that expression of a SIM-less mutant of SMN in a mouse model of SMA slightly extends survival rate by preventing motor neuron loss and slightly improving neuromuscular junction synapses, but w/o preventing the loss of sensory-motor synapses. The authors conclude that sumoylation is important for proper assembly and function of the SMN complex and that loss of SIM-Sumo interactions impaired the ability of SMN to correct specific deficits in the sensory-motor circuit of SMA mice.

This manuscript addresses a question likely of interest for the research community and readership of this journal: how does sumoylation affect the function of SMN in regulating RNP assembly and how does this affect the SMA disease process in animal models of SMA? It is well written and experimental data are generally of high quality. However, the first part of this manuscript partially overlaps with findings from Tapia et al., 2014, which is cited in this manuscript. In this paper the authors describe how the same SIM mutation impairs SMN interaction with SmD and coilin, and prevents CB assembly. Thus the novelty of this new manuscript lies in the focus on potentially sumoylated gemins, with Gemin5 being studied more closely as an example. There are major issues and open questions that should be addressed to increase the impact and novelty of the findings, as listed below:

1. This manuscript ignores a major role of sumoylation in regulating nucleocytoplasmic transport in background, results, and discussion: examples of protein sumoylation regulating nucleocytoplasmic transport have been documented, both through its effects on the properties of cargo molecules (e.g. PLK1 and many others), and by directly regulating the functions of components of the nuclear transport machinery (RanGAP1). Notably, depletion of UBC9 can be expected to abrogate the Ran-GTPase gradient regulated by sumoylated RanGAP1, interfering with nucleocytoplasmic transport. This alone can impact observed mislocalization and assembly defects and needs to be addressed. This is all the more important, since the authors claim a loss of SMN from the nucleus. Tapia et al conclude that SIM mutation affects gems/CB assembly and not nuclear import, but the imaging data are not very convincing. It would be important to show, if sumoylation regulates SMN nuclear import and interaction with importin-beta (Narayanan et al., “Coupled In Vitro Import of U snRNPs and SMN, the Spinal Muscular Atrophy Protein”. Mol. Cell 2004). This can be done e.g by biochemical nuclear vs. cytoplasmic fractionation and co-IP studies.

2. How are phenotypes correlated with SMA causing tudor domain mutations? E134K abrogates nuclear import (importin-interaction) and SmnD1 binding. Both these phenotypes have been observed for SIM mutants. It would be important to show whether E134K or other SMA causing mutations in the tudor domain affect SIM function in binding sumoylated gemins.

3. Are cytoplasmic foci similar to CBs in composition? Cytoplasmic “gems” would be more indicative of a nuclear import than assembly defect. Of course this would be predicted to affect RNP assembly function. Do mutated SMN and WT Gemin 5 or mutated Gemin5 and WT SMN colocalize in these? How does mutation of the sumoylation residues in Gemin 5 impact the protein’s localization? This would be important to show. It is also surprising that wild type Gemin 5 and SMN colocalize in cytoplasmic foci after UBC9 depletion, despite a proposed defect in binding (Suppl. Fig. 1).

Minor points:

“Previously, we have shown that motor neuron death in SMA is mediated by activation of the tumor suppressor p53, while sensory-motor circuit deficits are driven by dysfunction of Stasimon (Stas), a U12 intron-containing gene regulated by SMN.”

This has been reported for a specific mouse model of SMA, but not for SMA patients as stated. There

is also conflicting data with regard to p53 for the Smn2B⁻ mouse model.

Reviewer #2 (Remarks to the Author):

The manuscript by Riboldi and coworkers, analyzes the function of a putative SUMO Interactive Motif, SIM, located in the Tudor domain of the Survival Motor Neuron (SMN) protein. SMN accumulated in Gems/Cajal bodies and forms a complex with additional proteins (GEMIN2-8 and UNRIP) that has a role in the biogenesis of spliceosomal small nuclear ribonucleoproteins (snRNPs). Here, the authors show that mutations in the putative SIM sequence compromise the full function of SMN, mutant 2VA being unable to rescue all phenotypes produced by the absence of SMN in cells and animal model systems (zebrafish, mouse). The manuscript is in general well written. In general, the experiments are well controlled and conclusions derived from them are in most of the cases convincing. However, a number of concerns raised that preclude its publication from Nat Com in its present form.

- My main concern is about the nature of the SIM domain that is being analyzed here. It is not clear to me whether it has been demonstrated the capacity of this sequence for binding SUMO, either SUMO1 or SUMO2/3. Authors cite several times the publication by Tapia et al 2014 as a proof of the functionality of the SIM. However, I cannot see in that publication any convincing experiment that shows the capacity of the SIM sequence to bind SUMO. In fact, the Tapia et al say in the text “we cannot discard that the free groove of the covalently bound SUMO1 to SMN or that the SIM-like sequence mediate other molecular interactions.” As the functionality of that putative SIM domain is at the base of all the publication, the authors should first prove that this is an actual SIM.

- Supplementary Figure 3a shows that SMN-2VA mutant does not bind to GMIN4, which is not SUMOylated according to Figure 2, and binds weakly to GEMIN2, which is also not SUMOylated according to Figure 2. This will argue against the role of the putative sequence as a SUMO binding domain, but a domain important for protein-protein interaction.

- Mutation of the putative SIM domain in SMN impairs its ability to form large macromolecular complexes as shown in Figure 4a. However, this effect is stronger than the one shown in Figure 1e by silencing UBC9. This might argue against the only role of that sequence is as a SUMO binding.

- The most convincing experiment to show the function of the putative sequence as a SIM, is the one shown in Figure 4c, where Lysine mutations in GEMIN5 impair IP by SMN-2VA. However, this is an indirect prove that the interaction is mediated by the SUMO-SIM contact. A more direct evidence should be required, i.e. binding of the SIM sequence to SUMO1 or SUMO2/3.

- Title of Figure 4 is an overstatement, as it say “interaction with SUMO-modified components of the SMN complex”. Authors did not prove that the interaction is with the SUMO modified component, but interacts with a protein that can be SUMOylated. Also title of Fig. S3.

- The authors show that reducing SUMOylation by silencing UBC9, compromises the localization of the SMN complex components. However, this is not the SUMOylation of SMN itself, as the K mutant does not alter its location according to Tapia et al 2014 (Fig. 2D). So, might be the localization of other components of the complex. Authors should make this more clear in the text.

- The experiments of SUMOylation in vitro are convincing, and the authors make a number of in silico predictions. They should compare their predictions with the abundant published proteomics data on SUMOylated proteins (i.e. PMID: 27435506). Have the proteins of the SMN complex being found in SUMOylation screens? Do the sites found coincide with the ones predicted?

- The expression “non-covalent SMN sumoylation” is misleading. SUMOylation is defined as the covalent modification of a target protein by SUMO. The authors might be referring to a SUMO non-covalent interaction through a putative SIM region. If this is the case, use the term “SUMO interaction”.

- Figure 5b,c, the phenotypes should be specified to the non-specialist readership. Could you indicate in the Figure and explain in the text, which is the phenotype? Add examples of mild, severe and moderate.
- Figure 3C, bodies are not so evident, show the green channel separately, preferably in black and white.
- SMN 2VA has a shorter half life, but in all the transfections/transformations, they check levels are similar to WT. How this is explained?
- It is intriguing that righting reflex and weight gain improve during the first 10 days after injection and deteriorate thereafter. Could the authors give any mechanistic explanation for that?
- According to HUGO, the name of the gene should be SMN1, survival of motor neuron 1, telomeric. It might be more correct to refer to the protein as SMN1.
- Specify in Figure 7b, d) what is the phenotype to see (i.e. red buttons, lack of co-localization between SYP and BTX).
- Indicate in the Figure 1C the cytoplasmic foci.
- Figure Fig. 8a and 8c, not very visible, change color and zoom in to the phenotype.
- Page 7, Supplementary Figure 3, should be 2.
- In M&M, Triton X, specify, i.e. Triton X100.
- In M&M, you say that the statistical measurements are One Way ANOVA, but later in the figure legends you say T-test in several occasions.
- Fig. 4d, are those three independent experiments? Please specify
- Figs. 6A and 7D, indicate what the comparisons are, between which samples, i.e the asterisks close to the bars.
- Loading controls in WB, Fig 4b, c
- Scale bar in 7b does not seem right if the one in A is 25 micro m. Please revise.
- Fig. 9a, indicate which one is the aberrant form.
- Fig. S2b, specify the tags of the constructs

Below, we provide a point-by-point response to the reviewers' comments and outline the changes that we have made to the manuscript in response to the reviewers' concerns. We have reported all changes in the manuscript text file in a different color (red).

Reviewer #1 (Remarks to the Author):

In the first part of the manuscript "Sumoylation regulates the assembly and activity of the SMN complex", the authors show that mutating the SUMO-interacting motif (SIM) in SMN alters its sub-cellular distribution, SMN/Gemin complex assembly, and its function in snRNP biogenesis. The second part is more focused on animal models. The authors found that expression of a SIM-less mutant of SMN in a mouse model of SMA slightly extends survival rate by preventing motor neuron loss and slightly improving neuromuscular junction synapses, but w/o preventing the loss of sensory-motor synapses. The authors conclude that sumoylation is important for proper assembly and function of the SMN complex and that loss of SIM-Sumo interactions impaired the ability of SMN to correct specific deficits in the sensory-motor circuit of SMA mice. This manuscript addresses an question likely of interest for the research community and readership of this journal: how does sumoylation affect the function of SMN in regulating RNP assembly and how does this affect the SMA disease process in animal models of SMA? It is well written and experimental data are generally of high quality. However, the first part of this manuscript partially overlaps with findings from Tapia et al., 2014, which is cited in this manuscript. In this paper the authors describe how the same SIM mutation impairs SMN interaction with SmD and coilin, and prevents CB assembly. Thus the novelty of this new manuscript lies in the focus on potentially sumoylated gemins, with Gemin5 being studied more closely as an example. There are major issues and open questions that should be addressed to increase the impact and novelty of the findings, as listed below:

We thank this Reviewer for finding our data interesting and of high quality. We would like to point out that while Tapia *et al.* (2014) reported that mutations in the SIM-like domain of SMN disrupt its interaction with SmD1 and prevents formation of Cajal bodies, our work is the first one to investigate the role of these SUMO-SMN interactions in functional assays and in the disease context in two animal models of SMA. In addition, thanks to the suggestions of the

reviewers, we now have strong evidence that the SIM-like domain of SMN is able to bind to SUMO (Figure 3b), a novel aspect that was not addressed in Tapia *et al.*, (2014).

1. This manuscript ignores a major role of sumoylation in regulating nucleocytoplasmic transport in background, results, and discussion: examples of protein sumoylation regulating nucleocytoplasmic transport have been documented, both through its effects on the properties of cargo molecules (e.g. PLK1 and many others), and by directly regulating the functions of components of the nuclear transport machinery (RanGAP1). Notably, depletion of UBC9 can be expected to abrogate the Ran-GTPase gradient regulated by sumoylated RanGAP1, interfering with nucleocytoplasmic transport. This alone can impact observed mislocalization and assembly defects and needs to be addressed.

We thank the Reviewer for the opportunity to better detail the critical role of sumoylation in regulating nucleocytoplasmic transport. We have included this important function of sumoylation in Introduction (lines 79-87), Results (lines 129-135) and Discussion (lines 432-439).

This is all the more important, since the authors claim a loss of SMN from the nucleus. Tapia *et al* conclude that SIM mutation affects gems/CB assembly and not nuclear import, but the imaging data are not very convincing. It would be important to show, if sumoylation regulates SMN nuclear import and interaction with importin-beta (Narayanan *et al.*, “Coupled In Vitro Import of U snRNPs and SMN, the Spinal Muscular Atrophy Protein”. *Mol. Cell* 2004). This can be done e.g by biochemical nuclear vs. cytoplasmic fractionation and co-IP studies.

As proposed by this Reviewer, we sought to address this point by performing nuclear-cytoplasmic fractionation studies using both HeLa cells with UBC9 RNAi and NIH-3T3 cells expressing WT or SIM-less SMN. As anticipated by the Reviewer, our results in HeLa-UBC9^{RNAi} cells show that inhibition of sumoylation results in more SMN localizing in the cytoplasm, with concomitant reduction of nuclear SMN content (Suppl. Figure 1d and 1e). Strikingly, the same experiment performed in NIH-3T3 with endogenous SMN replaced by either WT or SIM-less (2VA) SMN shows that SMN-2VA subcellular distribution is not impaired, despite its inability to form gems (Suppl. Figure 4g and 4h). Collectively, these results suggest that the impaired nuclear import (and localization) of SMN in HeLa-UBC9^{RNAi} cells is the result of global sumoylation inhibition, while SMN nuclear import is not affected by the loss of SUMO interaction, consistent with what was reported in Tapia *et al.*, (2014).

2. How are phenotypes correlated with SMA causing tudor domain mutations? E134K abrogates nuclear import (importin-interaction) and SmnD1 binding. Both these phenotypes have been observed for SIM mutants. It would be important to show whether E134K or other SMA causing mutations in the tudor domain affect SIM function in binding sumoylated gemins.

To address this question, we tested SMA-causing mutation of SMN-E134K for its interaction with other core components of the SMN complex (Figure 3c). Contrary to SIM-less SMN, E134K mutation in SMN does not impair its interaction with GEMIN3 and GEMIN5, confirming that the SIM domain of SMN is required for its interaction with GEMIN3 and GEMIN5 (see lines 192-194 and lines 442-445).

3. Are cytoplasmic foci similar to CBs in composition? Cytoplasmic “gems” would be more indicative of a nuclear import than assembly defect. Of course this would be predicted to affect RNP assembly function. Do mutated SMN and WT Gemin 5 or mutated Gemin5 and WT SMN colocalize in these? How does mutation of the sumoylation residues in Gemin 5 impact the protein’s localization? This would be important to show. It is also surprising that wild type Gemin 5 and SMN colocalize in cytoplasmic foci after UBC9 depletion, despite a proposed defect in binding (Suppl. Fig. 1).

We thank the Reviewer for this insightful comment. We believe that cytoplasmic foci are aberrant aggregates of proteins including components of the SMN complex. They could be the result of loss of sumoylation in other proteins and maybe also the effect of nuclear import defects due to reduced sumoylation of key nuclear import players (as pointed by this Reviewer in point #1 and showed in Suppl. Fig. 1d and 1e). In addition, mutated SMN (2VA in Figure 4b) or mutated GEMIN5 when expressed on a WT background (G5-2KR in Figure 3e) do not show foci in the cytoplasm, but rather an almost complete loss of gems in the nucleus. These data are now included in the new version of the manuscript (see lines 206-208 and lines 241-243). We also believe that the fact that SMN and GEMIN5 co-localize in such cytoplasmic structure does not necessarily mean that they are functionally interacting. It is also possible that these structures represent aberrant aggregations of components of the SMN complex and that they are in close proximity in these structures but they have loose binding in the rest of the cytoplasm, as suggested by the reduced interaction of SMN with GEMIN3 and GEMIN5 upon UBC9 silencing (Suppl. Fig. 1f).

Minor points:

“Previously, we have shown that motor neuron death in SMA is mediated by activation of the tumor suppressor p53, while sensory-motor circuit deficits are driven by dysfunction of Stasimon (Stas), a U12 intron-containing gene regulated by SMN.”

This has been reported for a specific mouse model of SMA, but not for SMA patients as stated. There is also conflicting data with regard to p53 for the *Smn2B*^{-/-} mouse model.

We have changed our original sentence to “in a mouse model of SMA...” (lines 69-70) to account for the fact that we have not reported the activation of p53 in SMA patients and that motor neuron death in a milder model of the disease does not seem to depend on p53 activation.

Reviewer #2 (Remarks to the Author):

The manuscript by Riboldi and coworkers, analyzes the function of a putative SUMO Interactive Motif, SIM, located in the Tudor domain of the Survival Motor Neuron (SMN) protein. SMN accumulated in Gems/Cajal bodies and forms a complex with additional proteins (GEMIN2-8 and UNRIP) that has a role in the biogenesis of spliceosomal small nuclear ribonucleoproteins (snRNPs). Here, the authors show that mutations in the putative SIM sequence compromise the full function of SMN, mutant 2VA being unable to rescue all phenotypes produced by the absence of SMN in cells and animal model systems (zebrafish, mouse). The manuscript is in general well written. In general, the experiments are well controlled and conclusions derived from them are in most of the cases convincing. However, a number of concerns raised that preclude its publication from Nat Com in its present form.

We thank this Reviewer for finding our experiments well controlled and our conclusions convincing and his/her useful suggestions that have strengthened our study.

- My main concern is about the nature of the SIM domain that is being analyzed here. It is not clear to me whether it has been demonstrated the capacity of this sequence for binding SUMO, either SUMO1 or SUMO2/3. Authors cite several times the publication by Tapia et al 2014 as a proof of the functionality of the SIM. However, I cannot see in that publication any convincing experiment that shows the capacity of the SIM sequence to bind SUMO. In fact, the Tapia et al say in the text “we cannot discard that the free groove of the covalently bound SUMO1 to SMN or that the SIM-like sequence mediate other molecular interactions.” As the functionality of that putative SIM domain is at the base of all the publication, the authors should first prove that this is an actual SIM.

We agree with the Reviewer that Tapia *et al.* (2014) has not directly tested whether the SIM of SMN is able to bind SUMO. We assessed this interaction by incubating a GST-fused Tudor domain of SMN either WT or mutated (SIM-less) with SUMO-1 and SUMO-2/3. Then we used GST as a bait to pull down the SMN Tudor domain and assess whether it binds to SUMO. We found that the Tudor domain of SMN interacts preferentially with SUMO-2/3 and that this interaction is lost by mutation that abolish the SIM domain (Figure 3b).

- Supplementary Figure 3a shows that SMN-2VA mutant does not bind to GMIN4, which is not SUMOylated according to Figure 2, and binds weakly to GEMIN2, which is also not SUMOylated according to Figure 2. This will argue against the role of the putative sequence as a SUMO binding domain, but a domain important for protein-protein interaction.

We thank the Reviewer for pointing this out. It is not surprising that GEMIN4 binding is reduced because it is known that GEMIN4 does not bind directly to SMN, but through its interactions with GEMIN3 and GEMIN5 which bind directly to SMN (Battle *et al.*, *JBC* 2007; PMID: 17640873). As for GEMIN2, there is evidence that its binding to SMN is facilitated by the interaction with GEMIN5 (Otter *et al.*, *JBC* 2007; PMID: 17178713). Therefore, we believe that loss of SMN-GEMIN5 (and potentially GEMIN3) interaction might have an impact also on other non-sumoylated Gemins which interact with SMN through their binding to sumoylatable Gemins (i.e. GEMIN5 and GEMIN3).

- Mutation of the putative SIM domain in SMN impairs its ability to form large macromolecular complexes as shown in Figure 4a. However, this effect is stronger than the one shown in Figure 1e by silencing UBC9. This might argue against the only role of that sequence is as a SUMO binding.

We believe that the difference on the magnitude of the effect on SMN ability to form macromolecular complexes observed with the two cell systems is due to the combination of the following two factors:

1) UBC9 RNAi might be incomplete and some residual sumoylation of Gemin5 (and potentially of other sumoylated Gemins) might render the phenotype less severe than the mutant SMN-2VA which loses all SUMO interactions. In line with this interpretation, our new co-IP data of SMN in Hela-UBC9_{RNAi} cells show some residual binding to GEMIN5 and GEMIN3 (Suppl. Figure 1d);

2) In all experiments reported in Figure 1, we are looking at the effects of UBC9 silencing on the endogenous SMN complex, while in Figure 4 we are looking only at the mutated SMN-2VA with an antibody blind for the endogenous SMN protein. Therefore, the effect is expected to be more evident when using SMN-2VA.

- The most convincing experiment to show the function of the putative sequence as a SIM, is the one shown in Figure 4c, where Lysine mutations in GEMIN5 impair IP by SMN-2VA. However, this is an indirect prove that the interaction is mediated by the SUMO-SIM contact. A more direct evidence should be required, i.e. binding of the SIM sequence to SUMO1 or SUMO2/3.

To address this question, we performed new experiments which show an efficient interaction of the SIM domain of SMN with SUMO-2/3 (Figure 3b) and demonstrate that GEMIN-5 is modified by SUMO-2/3 (Supplementary Fig. 2). Thus, these new results provide more convincing evidence that SMN-GEMIN5 interaction is mediated (at least in part) by SUMO-SIM contacts.

- Title of Figure 4 is an overstatement, as it say "interaction with SUMO-modified components of the SMN complex". Authors did not prove that the interaction is with the SUMO modified component, but interacts with a protein that can be SUMOylated. Also title of Fig. S3.

Despite the fact that now we have more evidence indicating that SUMO-SIM contact is important for SMN-GEMIN5 interaction, we agree with this Reviewer that we have not directly proved that GEMIN5 is sumoylated when it interacts with SMN. Therefore, we have changed the original title of Figure 4 (now Figure 3) and Supplementary Fig. 3 in "The SIM domain of SMN is required for its interaction with SUMO modifiable components of the SMN complex."

- The authors show that reducing SUMOylation by silencing UBC9, compromises the localization of the SMN complex components. However, this if not the SUMOylation of SMN itself, as the K mutant does not alter its location according to Tapia et al 2014 (Fig. 2D). So, might be the localization of other components of the complex. Authors should make this more clear in the text.

The Reviewer is correct that silencing UBC9 has a wider effect, including an effect on nucleocytoplasmic trafficking of the SMN complex which has been assessed with the nuclear-cytoplasmic fractionation studies in Hela-UBC9_{RNAi} cells (see also answer to Reviewer #1, point #1). We have added a sentence in Results (lines 132-135) and Discussion (lines 432-439) to make this point clearer.

- The experiments of SUMOylation in vitro are convincing, and the authors make a number of in silico predictions. They should compare their predictions with the abundant published

proteomics data on SUMOylated proteins (i.e. PMID: 27435506). Have the proteins of the SMN complex being found in SUMOylation screens? Do the sites found coincide with the ones predicted?

We appreciate the Reviewer's comment on the quality of the *in vitro* sumoylation experiments. As suggested, we reviewed the published proteomics data relative to the sumoylation of the core components of the SMN complex. We considered the work this Reviewer suggested (Hendriks *et al.*, *Nat Rev Mol Cell Biol*, 2016; PMID: 27435506), which remains the most comprehensive proteomics data repository on sumoylation proteomics screening, as well as the database <http://proteomecentral.proteomexchange.org/cgi/GetDataset>. Datasets were filtered using the searching term "sumoylation". We identified 26 additional papers. We compared the predicated data of sumoylation of the SMN complex core components (as reported in Supplementary Table 1) and the ones reported in Hendriks *et al.*, 2016. The overlaps between proteomics data and the bioinformatically predicted sites are detailed in Supplementary Table 2.

- The expression "non-covalent SMN sumoylation" is misleading. SUMOylation is defined as the covalent modification of a target protein by SUMO. The authors might be referring to a SUMO non-covalent interaction through a putative SIM region. If this is the case, use the term "SUMO interaction".

We thank this Reviewer for pointing this out and apologize if we were misleading in our use of the term "non-covalent sumoylation." In agreement with Reviewer's suggestion, we changed "non-covalent SMN sumoylation" to "SUMO-SIM interaction" in the text.

- Figure 5b,c, the phenotypes should be specified to the non-specialist readership. Could you indicate in the Figure and explain in the text, which is the phenotype? Add examples of mild, severe and moderate.

We thank the Reviewer for the opportunity to clarify the phenotypes in the text. We modified the figure and the legend to make the phenotype clearer and more immediate.

- Figure 3C, bodies are not so evident, show the green channel separately, preferably in black and white.

We have changed the original Figure 3c (now Figure 4b) with a new representative image of the same cells in which gems more evident. As per Reviewer's suggestion we are reporting the image in black and white.

- SMN 2VA has a shorter half-life, but in all the transfections/transformations, they check levels are similar to WT. How this is explained?

This is an excellent point as it is true that the faster turnover of the SMN-2VA protein relative to SMN-WT results in slightly reduced steady-state levels of the mutant protein. This discrepancy is due to the fact that SMN-2VA stability is influenced by the presence of endogenous SMN, therefore its real turnover can only be measured when endogenous SMN is depleted (as we have done in Figure 4d and 4e). In these conditions SMN-2VA steady-state protein levels are less than SMN-WT even if their transcript levels are similar. To make this point clearer we now show a direct comparison and quantification of SMN-WT and SMN-2VA levels in the presence and absence of endogenous SMN (Suppl. Figure 4d and 4e). In this comparison it is clear that SMN-2VA steady-state levels are slightly less than SMN-WT levels in the presence of endogenous SMN (-Dox) and that SMN-2VA levels are further reduced upon induction of endogenous SMN silencing (+Dox). We have also changed the representative Western blots on Fig. 4d to better reflect the different steady-state levels and turnover of SMN-2VA.

- It is intriguing that righting reflex and weight gain improve during the first 10 days after injection and deteriorate thereafter. Could the authors give any mechanistic explanation for that?

We thank the Reviewer for this intriguing question. The righting reflex is due to the proper function of spinal motor neurons, proprioceptive sensory neurons, spinal interneurons and the descending vestibule-spinal pathways. During normal development in the first postnatal week, the descending pathways are immature and have a small role, likely reflecting the prominence of proprioceptive neurons and motor neurons as we have previously shown (Fletcher *et al.*, *Nat Neurosci*, 2017; PMID: 28504671). During the second postnatal week, the vestibule-spinal pathways are likely more dominant in regulating the righting reflex. Thus, the transient improvement is expected for a partial suppression of the SMA phenotype, likely affecting the descending pathways on the motor circuit. To this end, metabolic disturbances during the second postnatal week that become more prominent and are not affected by our intervention may be responsible for the loss of weight gain. A similar situation is observed in studies that have attempted correction of the SMA phenotype with downstream targets of SMN that result in selective rescue of only certain functions (Simon *et al.*, *Cell Rep.*, 2019; PMID: 31851921).

- According to HUGO, the name of the gene should be SMN1, survival of motor neuron 1, telomeric. It might be more correct to refer to the protein as SMN1.

Both SMN1 and SMN2 genes contribute to the total levels of SMN protein, for this reason “SMN” is used when referring to the protein.

- Specify in Figure 7b, d) what is the phenotype to see (i.e. red buttons, lack of co-localization between SYP and BTX).

As suggested by the Reviewer, we added an explanation of the phenotype in the legend and indicated denervated neuromuscular junctions with white arrowheads in the figure.

- Indicate in the Figure 1C the cytoplasmic foci.

As suggested by the Reviewer, we added white arrowheads in the figure to indicate the cytoplasmic foci.

- Figure Fig. 8a and 8c, not very visible, change color and zoom in to the phenotype.

We thank this Reviewer for the comment. To make the figure clearer, we added an explanation of the phenotype in the figure legend and in the text (lines 356-357).

- Page 7, Supplementary Figure 3, should be 2.

We apologize for this oversight, which has now been corrected in the revised version of the manuscript.

- In M&M, Triton X, specify, i.e. Triton X100.

This missing detail has been corrected in the revised version of the manuscript.

- In M&M, you say that the statistical measurements are One Way ANOVA, but later in the figure legends you say T-test in several occasions.

We apologize for this omission. We revised the *Statistical analysis* section of *Methods* to include all statistical tests and methods used in this study (see lines 771-783).

- Fig. 4d, are those three independent experiments? Please specify

The experiments reported in the original Fig. 4d (now Fig. 4f) are from three independent experiments. This is now reported in the legend of Fig. 4.

- Figs. 6A and 7D, indicate what the comparisons are, between which samples, i.e the asterisks close to the bars.

We would like to apologize if the comparisons in the cited figures were not clear. In the figure legends and in the *Statistical analysis* section of *Methods* we indicated what comparison the asterisks close to the bars refer to (see lines 779-781).

- Loading controls in WB, Fig 4b, c

We added loading control (alpha-tubulin) to the WBs in the original Fig. 4b and 4c (now Fig. 3c and 3d), and to the Supplementary Fig. 4f.

- Scale bar in 7b does not seem right if the one in A is 25 micro m. Please revise.

We thank the Reviewer for pointing this out. We revised all scale bars and corrected accordingly.

- Fig. 9a, indicate which one is the aberrant form.

We indicated the aberrant splicing of *Stasimon* in the figure legend (the red lines show the aberrant splicing form).

- Fig. S2b, specify the tags of the constructs

Per Reviewer's suggestion, we specified the tags of the construct in all figures.

In light of the above, we believe that our revised manuscript fully addresses the Reviewers' comments. Moreover, thanks to the constructive comments from the Reviewers, we were able to significantly improve and strengthen our manuscript. We are looking forward to hearing your views on our revised manuscript.

Francesco Lotti

REVIEWER COMMENTS

Reviewer #1 (Remarks to the Author):

The reviewers have now addressed this reviewer's concerns and considerably strengthened the manuscript.

Remaining minor issues are listed below:

The title implies that sumoylation is the one process governing assembly and activity. I would suggest e.g. "The assembly and activity of the SMN complex is regulated by sumoylation".

The revised text would benefit from thorough editing for clarity. A few examples are listed below:

The text mentions repeatedly "loss of a SUMO-interacting motif (SIM)" and "SIM-less mutant of SMN" etc. This implies a deletion of the SIM despite experiments using a mutated SIM. Better would be "mutation (or inactivation) of a SUMO-interacting motif (SIM)" and "Expression of SMN with a mutant SIM" etc.

In line with this possibility, we found that SMN interaction with GEMIN3 and GEMIN5 is weakened by inhibition of sumoylation (Supplementary Fig. 1f). -> is weakened

In addition to SUMO-1, SMN, GEMIN3 and GEMIN5 are also modified by SUMO-2 (Supplementary Fig. 2). -> In addition to modification with SUMO-1, SMN.....

Overall, these results indicate that SUMO-SIM contacts are important for SMN complex localization and stability. -> ...that SUMO-SIM binding is important....

Figures use a,b,c,d while figure legends use capitalized A,B,C,D.

Reviewer #2 (Remarks to the Author):

The authors responded satisfactorily to the Reviewers requests. As they show now in Figure 2B, they proved the functionality of the SIM domains in the protein.

Below, we provide a point-by-point response to the reviewers' comments and outline the changes that we have made to the manuscript in response to the reviewers' comments.

Reviewer #1 (Remarks to the Author):

Remaining minor issues are listed below:

The title implies that sumoylation is the one process governing assembly and activity. I would suggest e.g. "The assembly and activity of the SMN complex is regulated by sumoylation".

In our opinion, the present title better conveys the results of the paper. This Reviewer suggested the use of a passive form, which we think may be less efficacious in delivering the message; however, we will follow the suggestion of the editor on this matter.

The text mentions repeatedly "loss of a SUMO-interacting motif (SIM)" and "SIM-less mutant of SMN" etc. This implies a deletion of the SIM despite experiments using a mutated SIM. Better would be "mutation (or inactivation) of a SUMO-interacting motif (SIM)" and "Expression of SMN with a mutant SIM" etc.

We changed "SIM-less" to "SIM-inactivated" throughout the revised text.

In line with this possibility, we found that SMN interaction with GEMIN3 and GEMIN5 is weakened by inhibition of sumoylation (Supplementary Fig. 1f). -> is weakened

We corrected this typo.

In addition to SUMO-1, SMN, GEMIN3 and GEMIN5 are also modified by SUMO-2 (Supplementary Fig. 2). -> In addition to modification with SUMO-1, SMN.....

We modified the text as suggested.

Overall, these results indicate that SUMO-SIM contacts are important for SMN complex localization and stability. -> ...that SUMO-SIM binding is important....

We modified the text as suggested.

Figures use a,b,c,d while figure legends use capitalized A,B,C,D.

We modified the figure legends using small letters.

Reviewer #2 (Remarks to the Author):

The authors responded satisfactorily to the Reviewers requests. As they show now in Figure 2B, they proved the functionality of the SIM domains in the protein.

We thank this Reviewer for finding our revision satisfactory.